# Local Linear Convergence of Projected Gradient Descent: A Discrete and Continuous Analysis

## Abstract

Projected Gradient Descent (PGD) method has been successfully applied to various machine learning problems. The prior work Nesterov (2018) demonstrates that PGD, as a classical discrete iterative method, has sublinear convergence in the case that the objective function is convex and smooth. In this paper, we explore the local linear convergence properties of PGD under this case from both discrete and continuous perspectives. Specifically, we focus on optimization problems with a general convex and smooth objective function constrained by $\mathbb{B}(\mathbf{0}, \epsilon)$, and present the following principal results: **(I)** We derive an ordinary differential equation (ODE) that arises as the limit of PGD; **(II)** We establish convergence rate bounds for PGD in both discrete-time and continuous-time scenarios, with the continuous-time analysis motivated by the derived ODE. The bounds in both scenarios support each other and consistently indicate that PGD achieves a local linear convergence rate. Finally, we conduct experiments to validate theoretical results **(I)** and **(II)**, and the experimental outcomes closely align with our theoretical findings.

## 1 Introduction

Constrained optimizations play a critical role in many fields of machine learning (Amini et al., 2025; Beznosikov et al., 2024; Shalev-Shwartz & Ben-David, 2014). As highlighted in prior works (Madry et al., 2018; Nesterov, 2018; Deng et al., 2020; Lee et al., 2024; Wang et al., 2024), the constraint $\mathbb{B}(\mathbf{0}, \epsilon)$ is frequently employed in constrained optimization problems extracted from machine learning tasks. In the simplest and most standard form, the optimization problem (OP) is defined as:

$$\min_{\boldsymbol{\delta} \in \mathbb{B}(\mathbf{0}, \epsilon)} \Phi(\boldsymbol{\delta}), \tag{1}$$

where $\Phi(\cdot) : \mathbb{R}^d \to \mathbb{R}$ is a general convex and smooth function, $\boldsymbol{\delta} = (\delta^{(1)}, \delta^{(2)}, \cdots, \delta^{(d)}) \in \mathbb{R}^d$, $\|\cdot\|$ represents the standard Euclidean norm ($l_2$ norm), and $\mathbb{B}(\mathbf{0}, \epsilon) := \{\boldsymbol{\delta} \mid \|\boldsymbol{\delta} - \mathbf{0}\| \leq \epsilon\}$ with $\epsilon \in \mathbb{R}$. Let $\boldsymbol{\delta}^{\triangle}$ be the optimal solution to OP (1), i.e., $\boldsymbol{\delta}^{\triangle} = \arg\min_{\|\boldsymbol{\delta}\| \leq \epsilon} \Phi(\boldsymbol{\delta})$. Moreover, let $\boldsymbol{\delta}^*$ represent the equilibrium point of $\Phi(\boldsymbol{\delta})$, defined by the condition $\frac{\partial \Phi(\boldsymbol{\delta})}{\partial \boldsymbol{\delta}}|_{\boldsymbol{\delta}=\boldsymbol{\delta}^*} = \mathbf{0}$.

Various methods have been proposed to solve OP (1), such as the interior point method (Gill & Zhang, 2024; Domahidi et al., 2012), the Lagrange multiplier method (Haeser et al., 2021), the projected gradient descent (PGD) method (Bubeck, 2015), and so on. Among them, the PGD method achieves significant success (Bryniarski et al., 2022; Bloom et al., 2016). In accordance with works (Nesterov, 2018; Boyd & Vandenberghe, 2014; Madry et al., 2018), the standard PGD method for solving OP (1) is a classical discrete iterative approach formulated as follows:

$$\begin{cases} \widetilde{\boldsymbol{\delta}}_{k+1} = \boldsymbol{\delta}_k - \eta \nabla \Phi(\boldsymbol{\delta}_k) \\ \boldsymbol{\delta}_{k+1} = \mathcal{P}_{\mathbb{B}(\mathbf{0}, \epsilon)}[\widetilde{\boldsymbol{\delta}}_{k+1}] \end{cases}, \quad k \geq 0, \tag{2}$$

where $\mathcal{P}$ denotes the projection operator, defined for all $\widetilde{\boldsymbol{\delta}} \in \mathbb{R}^d$, $\mathcal{P}_{\mathbb{B}(\mathbf{0}, \epsilon)}[\widetilde{\boldsymbol{\delta}}] = \arg\min_{\boldsymbol{\delta} \in \mathbb{B}(\mathbf{0}, \epsilon)} \|\boldsymbol{\delta} - \widetilde{\boldsymbol{\delta}}\|$. Besides, $\eta$ represents the step size, and $\boldsymbol{\delta}_0$ is the initial point sampled from $\mathbb{B}(\mathbf{0}, \epsilon)$.

According to previous works (Nesterov, 2018; Boyd & Vandenberghe, 2014; Bubeck, 2015; Necoara et al., 2019), when the objective function $\Phi(\boldsymbol{\delta})$ is both convex and smooth, the sequence $\boldsymbol{\delta}_k$ generated

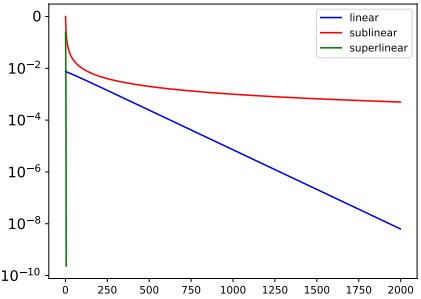

Figure 1: Comparison of sublinear, linear, and superlinear convergence. The red line represents $k^{-1}$, indicating sublinear convergence. The green line represents $2^{-2^k}$, demonstrating superlinear convergence. The blue straight line corresponds to the linear convergence of PGD when solving OP (1) with $\Phi(\boldsymbol{\delta}) = \delta^{(1)} + \delta^{(2)}$ under the constraint $\boldsymbol{\delta} \in \mathbb{B}(\mathbf{0}, 0.01)$. The initialization point $\boldsymbol{\delta}_0$ is sampled uniformly from $\{\boldsymbol{\delta}|\boldsymbol{\delta} \in \mathbb{S}(\mathbf{0}, 0.01), \delta^{(1)} < 0, \delta^{(2)} < 0\}$, and the sequence $\boldsymbol{\delta}_k$ are generated by PGD (2). The optimal point is $\boldsymbol{\delta}^{\triangle} = (\frac{-\sqrt{2}}{200}, \frac{-\sqrt{2}}{200})^{\top}$. A logarithmic coordinate system is used, where the horizontal axis represents the number of iterations $k$ and the vertical axis represents $\log_{10}(\|\boldsymbol{\delta}_k - \boldsymbol{\delta}^{\triangle}\|)$.

by PGD is $O(\frac{1}{k})$ close to the optimal point $\boldsymbol{\delta}^{\triangle}$ within simple sets (including Euclidean balls) at the $k$-th iteration. This implies that the convergence rate of PGD for solving OP (1) is sublinear. However, as illustrated by the blue straight line in Figure 1, when $\Phi(\boldsymbol{\delta})$ is a linear function, PGD exhibits a linear convergence rate. Since linear functions are smooth and convex, this suggests that for general convex and smooth objective functions, PGD can achieve linear rather than sublinear convergence in certain situations when solving OP (1). As demonstrated in recent studies (Vu et al., 2023; Fiez & Ratliff, 2021; Poon et al., 2018; Wang & Chizat, 2023), the analysis of local convergence has emerged as a significant and widely recognized area of focus within optimization research. Furthermore, to establish the convergence of discrete iterative methods, the work presented in (Su et al., 2016) modeled Nesterov's accelerated gradient method as an ODE. Building on these insights, this paper conducts a detailed investigation into the local linear convergence properties of PGD (2) for solving OP (1) from both discrete-time and continuous-time perspectives.

Because PGD aligns with the gradient descent method when $\boldsymbol{\delta}^* \in \mathbb{B}(\mathbf{0}, \epsilon)$, its convergence rate for solving OP (1) is sublinear for a general smooth and convex objective function (Nesterov, 2018; Boyd & Vandenberghe, 2014; Bubeck, 2015). Consequently, this paper is primarily centered on the case that there is no $\boldsymbol{\delta}^*$ in $\mathbb{B}(\mathbf{0}, \epsilon)$. To simplify the analysis, we focus on the local convergence behavior of PGD around $\boldsymbol{\delta}^{\triangle}$, where the definition of local convergence only requires the existence of such a neighborhood (Nesterov, 2018). When $\boldsymbol{\delta}^* \notin \mathbb{B}(\mathbf{0}, \epsilon)$, the evolutionary processes of the PGD algorithm are depicted in Figure 2. As illustrated in Figure 2, we have $\boldsymbol{\delta}^{\triangle} \in \mathbb{S}(\mathbf{0}, \epsilon)$, where $\mathbb{S}(\mathbf{0}, \epsilon) := \{\boldsymbol{\delta}| \|\boldsymbol{\delta} - \mathbf{0}\| = \epsilon\}$. The convergence analysis of PGD for solving OP (1) is conducted from both discrete-time and continuous-time perspectives, corresponding to Figures 2 (a) and 2 (b), respectively.

**In the discrete-time scenario**, as depicted in Figure 2 (a), PGD (2) is directly applied to solve OP (1). We establish both upper and lower bounds for the convergence rate of PGD (2). Specifically, the sequence $\boldsymbol{\delta}_k$ generated by PGD (2) is $O(C_1^k)$ (where $0 < C_3 < C_1 < C_2 < 1$) close to the optimal point $\boldsymbol{\delta}^{\triangle}$ at the $k$-th iteration, where $C_1, C_2, C_3$ are constants, with $C_2$ determining the upper bound and $C_3$ characterizing the lower bound of the convergence rate.

**In the continuous-time scenario**, as depicted in Figure 2 (b), we derive an ODE that represents the limiting behavior of PGD (2) as the step size approaches zero. This ODE is formulated as:

$$\begin{cases} \dfrac{\partial \boldsymbol{\delta}(t)}{\partial t} = \dfrac{1}{\epsilon^2}\boldsymbol{\delta}\langle \boldsymbol{\delta}, \nabla\Phi(\boldsymbol{\delta})\rangle - \nabla\Phi(\boldsymbol{\delta}), \\ \boldsymbol{\delta}(0) = \boldsymbol{\delta}_0, \ t \geq 0, \end{cases} \tag{3}$$

where $\boldsymbol{\delta}(t) : [0, +\infty) \to \mathbb{R}^d$. The time parameter $t$ in this ODE is associated with the step size $\eta$ in PGD (2) via the relationship $t_k = k\eta$. The ODE (3) demonstrates approximate equivalence to PGD (2) and thus offers a continuous-time perspective. This differs from the classical discrete-time formulation of PGD studied by papers (Nesterov, 2018; Boyd & Vandenberghe, 2014; Bubeck, 2015). We rigorously prove that the optimal point $\boldsymbol{\delta}^{\triangle}$ of OP (1) is the exponential asymptotical equilibrium

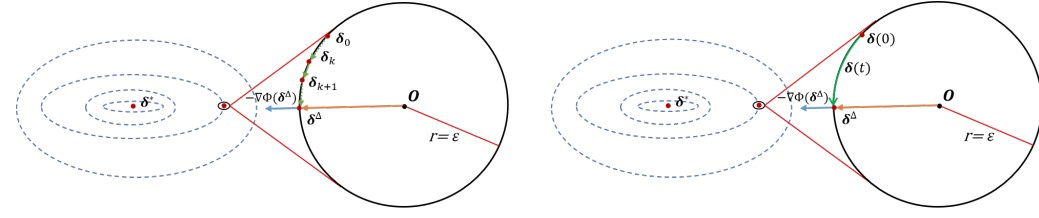

(a) Trajectory generated by PGD.

(b) Trajectory determined by ODE.

Figure 2: Trajectories of PGD in discrete-time and continuous-time scenarios for $\boldsymbol{\delta}^* \notin \mathbb{B}(\mathbf{0}, \epsilon)$. The dashed blue lines are the contour lines of $\Phi(\boldsymbol{\delta})$, and the region enclosed by the bold black circle denotes $\mathbb{B}(\mathbf{0}, \epsilon)$. The ball $\mathbb{B}(\mathbf{0}, \epsilon)$ is significantly small, and $\boldsymbol{\delta}_0$ is in close proximity to the optimal point $\boldsymbol{\delta}^{\triangle}$. For clarity, we magnify the area where $\mathbb{B}(\mathbf{0}, \epsilon)$ located.

stable point of ODE (3). In other words, $\boldsymbol{\delta}(t)$ determined by ODE (3) remains $O(e^{C_4 t})$ (where $C_4 < 0$) close to the optimal point $\boldsymbol{\delta}^{\triangle}$ at evolutionary time $t$.

The theoretical results from both discrete-time and continuous-time scenarios support each other and consistently indicate that when the objective function $\Phi(\boldsymbol{\delta})$ is convex and smooth, the convergence rate of PGD for solving OP (1) is local linear, complementing the convergence results of prior works (Nesterov, 2018; Boyd & Vandenberghe, 2014; Bubeck, 2015).

We also perform numerical experiments to validate our theoretical findings. The experimental results confirm the following: **(I)** ODE (3) accurately represents the continuous-time limit of PGD (2); **(II)** The experimental results from both continuous-time and discrete-time scenarios mutually reinforce each other and consistently demonstrate that PGD achieves local linear convergence under the assumption of general convexity and smoothness for the objective function. These experimental results are in excellent agreement with our theoretical predictions.

## 2 PRELIMINARY AND OUTLINE

### 2.1 PRELIMINARY

Below, we summarize some standard notations used throughout the paper.

**Definition 2.1.** $\Phi(\boldsymbol{\delta})$ is $M$-smooth over $\mathbb{R}^d$, that is, there exists $M > 0$, $\|\nabla\Phi(\widetilde{\boldsymbol{\delta}}) - \nabla\Phi(\boldsymbol{\delta})\| \leq M\|\widetilde{\boldsymbol{\delta}} - \boldsymbol{\delta}\|$, $\forall \boldsymbol{\delta}, \widetilde{\boldsymbol{\delta}} \in \mathbb{R}^d$.

We use $\mathcal{F}_M$ to denote the class of convex functions $\Phi(\boldsymbol{\delta})$ that are $M$-smooth, i.e., $\Phi(\boldsymbol{\delta})$ is convex, continuously differentiable, and satisfies Definition 2.1. Additionally, $C^1([0, \infty); \mathbb{R}^d)$ refers to the class of continuously differentiable maps from $[0, \infty)$ to $\mathbb{R}^d$. Furthermore, $\mathbb{Z}^+$ denotes the set of positive integers. $E$ represents the identity matrix, and $I$ is a vector with all elements equal to 1. Finally, $\chi(\cdot)$ extracts the eigenvalues of its argument.

As shown in Figures 2, when $\Phi(\boldsymbol{\delta}) \in \mathcal{F}_M$ and $\boldsymbol{\delta}^* \notin \mathbb{B}(\mathbf{0}, \epsilon)$, $\mathbb{B}(\mathbf{0}, \epsilon)$ is externally tangential to a contour line of $\Phi(\boldsymbol{\delta})$ at $\boldsymbol{\delta}^{\triangle}$. Consequently, $\boldsymbol{\delta}^{\triangle} \in \mathbb{S}(\mathbf{0}, \epsilon)$, and $-\nabla\Phi(\boldsymbol{\delta}^{\triangle}) = \lambda\boldsymbol{\delta}^{\triangle}$ for some $\lambda > 0$ (Nesterov, 2004; Boyd & Vandenberghe, 2014). The case where $\boldsymbol{\delta}^* \notin \mathbb{B}(\mathbf{0}, \epsilon)$ also encompasses situations in which $\Phi(\boldsymbol{\delta})$ doesn't have $\boldsymbol{\delta}^*$.

To analyze PGD from a continuous-time perspective, we introduce the necessary tools for working with ODEs. The commonly used autonomous ODE is expressed as:

$$\partial\boldsymbol{\delta}(t)/\partial t = \boldsymbol{F}(\boldsymbol{\delta}), \ \boldsymbol{\delta}(0) = \boldsymbol{\delta}_0, \tag{4}$$

where $\boldsymbol{\delta}(t)$ represents the solution to ODE (4). The Lemma D.1 in Appendix D.3 ensures the existence and uniqueness of $\boldsymbol{\delta}(t)$.

ODE (4) can be solved numerically. Assuming $\boldsymbol{\delta}_k$ represents the numerical solution of ODE (4), the solution $\boldsymbol{\delta}(t)$ can be approximated as $\boldsymbol{\delta}_k \approx \boldsymbol{\delta}(k\eta)$, where $k \geq 0$ and $t \geq 0$ (Su et al., 2016). The core concept behind the numerical solutions is to approximate the ODE by converting it into a discrete difference equation through a discretization method (Kincaid et al., 2009). Single-step methods are pivotal in such discretization techniques (Sauer, 2011). The explicit single-step method for solving ODE (4) is given by:

$$\boldsymbol{\delta}_{k+1} = \boldsymbol{\delta}_k + \eta\varphi(t_k, \boldsymbol{\delta}_k, \eta), \tag{5}$$

where $\varphi(t, \boldsymbol{\delta}, \eta)$ is the incremental function, and $\eta$ denotes the time step size. Various single-step methods correspond to distinct forms of incremental function. For instance, employing the forward Euler method to address ODE (4), we have $\varphi(t, \boldsymbol{\delta}, \eta) = \boldsymbol{F}(\boldsymbol{\delta})$,

The transition between ODE (4) and the difference equation (5) is valid if the numerical solution $\boldsymbol{\delta}_k$ of the difference equation (5) converges to the exact solution $\boldsymbol{\delta}(t_k)$ of the ODE (4), where $t_k = k\eta$. Rigorously, the relationship between $\boldsymbol{\delta}_k$ and $\boldsymbol{\delta}(t_k)$ must satisfy the following definition (Kincaid et al., 2009):

**Definition 2.2.** The single-step method (5) is called convergent, if $\lim_{\eta \to 0} \boldsymbol{\delta}_k = \boldsymbol{\delta}(t_k)$ for $\forall t_k \in [0, T]$.

In this paper, we reformulate PGD (2) into the discrete form (5) and subsequently derive its corresponding ODE in the form of (4). Definition 2.2 is applied to verify the validity of the ODE, which is critical for establishing the equivalence between PGD and its continuous-time counterpart.

## 2.2 OUTLINE

The subsequent sections of this paper are organized as follows: Section 3 establishes the upper and lower bounds for the convergence rate of PGD in a discrete-time scenario. In Section 4, we derive ODE (3) from PGD (2) and demonstrate its correctness. Inspired by ODE (3), we further provide a convergence rate bound for PGD from a continuous-time perspective. Section 5 summarizes and interprets the connections between the theoretical results obtained in the discrete-time and continuous-time scenarios. Finally, in Section 6, we present numerical experiments to validate the theoretical findings. Related works are discussed in Appendix A, and the limitations of our work are detailed in Appendix B

## 3 THE DISCRETE-TIME SCENARIO

In this section, we analyze the local convergence rate of PGD in a discrete-time scenario. To simplify the analysis, we first focus on a specific neighborhood and later extend the results to a general neighborhood. When $\boldsymbol{\delta}^* \notin \mathbb{B}(\boldsymbol{0}, \epsilon)$ and $\Phi(\boldsymbol{\delta}) \in \mathcal{F}_M$, the trajectory of $\boldsymbol{\delta}_k$ generated by the discrete PGD (2) is depicted in Figure 2 (a). The upper and lower bounds for the convergence rate of PGD (2) in finding $\boldsymbol{\delta}^{\triangle}$ are presented below.

**Theorem 3.1** (Upper and Lower Bounds). *Suppose that (i) $\Phi(\boldsymbol{\delta}) \in \mathcal{F}_M$ and $\boldsymbol{\delta}^* \notin \mathbb{B}(\boldsymbol{0}, \epsilon)$, (ii) $\epsilon < \frac{\|\nabla \Phi(\boldsymbol{0})\|}{(U+1)M}$ with $U \geq 2$, (iii) $\eta < \frac{1}{2M}$, (iv) $\boldsymbol{\delta}_k$ is generated by the PGD (2). Then, there exists $\theta \in (0, \frac{\pi}{3})$, if $\boldsymbol{\delta}_0 \in \mathbb{S}(\boldsymbol{0}, \epsilon) \cap \mathbb{B}(\boldsymbol{\delta}^{\triangle}, 2\epsilon \sin \frac{\theta}{2})$, we have $\boldsymbol{\delta}_k \in \mathbb{S}(\boldsymbol{0}, \epsilon)$ for $k \in \mathbb{Z}^+$ and*

$$C_3^k \|\boldsymbol{\delta}_0 - \boldsymbol{\delta}^{\triangle}\| \leq \|\boldsymbol{\delta}_k - \boldsymbol{\delta}^{\triangle}\| \leq C_2^k \|\boldsymbol{\delta}_0 - \boldsymbol{\delta}^{\triangle}\|, \tag{6}$$

*where $C_2 = \frac{2+3\eta M}{2+2U\eta M} < 1$ and $C_3 = \frac{2-3\eta M}{2+2U\eta M} < C_2 < 1$.*

See proof in Appendix C.

*Remark* 3.2. Theorem 3.1 indicates that when $\boldsymbol{\delta}^* \notin \mathbb{B}(\boldsymbol{0}, \epsilon)$, the sequence $\boldsymbol{\delta}_k$ generated by PGD (2) is $O(C_1^k)$ close to $\boldsymbol{\delta}^{\triangle}$ at the $k$-th iteration, where $C_1 \in [C_3, C_2]$ with $0 < C_3 < C_2 < 1$. This establishes that the convergence rate of PGD (2) is locally linear. According to the proofs in Appendix C, Theorem 3.1 remains valid even if $\Phi(\boldsymbol{\delta})$ doesn't have $\boldsymbol{\delta}^*$. Additionally, to maintain local linear convergence, $\epsilon$ can be relaxed to $\frac{\|\nabla \Phi(\boldsymbol{0})\|}{3M}$.

*Remark* 3.3. Theorem 3.1 focuses on the convergence behavior of the PGD method within the specific neighborhood of $\boldsymbol{\delta}^{\triangle}$, denoted by $\mathbb{S}(\boldsymbol{0}, \epsilon) \cap \mathbb{B}(\boldsymbol{\delta}^{\triangle}, 2\epsilon \sin \frac{\theta}{2})$. Appendix C.4 further demonstrates that this local linear convergence property holds in a general neighborhood, and the selection condition for the initial point $\boldsymbol{\delta}_0$ can be relaxed to $\boldsymbol{\delta}_0 \in \mathbb{B}(\boldsymbol{0}, \epsilon)$. In other words, the PGD method exhibits both global convergence and local linear convergence.

## 4 THE CONTINUOUS-TIME SCENARIO

In this section, we analyze PGD in a continuous-time scenario. When $\boldsymbol{\delta}^* \notin \mathbb{B}(\boldsymbol{0}, \epsilon)$, the trajectory of $\boldsymbol{\delta}(t)$ determined by ODE (3) is depicted in Figure 2 (b). This section is arranged as follows: Section 4.1 derives ODE (3) from PGD (2) by reducing the time step size $\eta$. Subsequently, Section 4.2

verifies the correctness of ODE (3) through trajectory constraints, the existence and uniqueness of the solution, and the rigorous convergence from PGD to ODE. Utilizing ODE (3), Section 4.3 establishes the upper bound for the convergence rate of PGD in the continuous-time scenario.

## 4.1 DERIVATION FROM PGD TO ODE

The connection between ODEs and numerical optimization is often established by taking infinitesimal step sizes, leading the trajectory or solution path of numerical optimization to converge toward a curve modeled by an ODE (Su et al., 2016). The succinct and well-established theory of ODEs offers useful tools into optimization, enabling significant advancements (Weinan, 2017). This chapter identifies the ODE corresponding to PGD (2). We begin by presenting an informal derivation of ODE (3).

Under the conditions of Theorem 3.1, the trajectory of $\boldsymbol{\delta}_k$ lies on $\mathbb{S}(\mathbf{0}, \epsilon)$ (See Lemma C.1). Assuming $\boldsymbol{\delta}(t) \in C^1([0, \infty); \mathbb{R}^d)$ and $\boldsymbol{\delta}_k = \boldsymbol{\delta}(t_k)$ with $t_k = k\eta$, PGD (2) can be reformulated as:

$$\boldsymbol{\delta}_{k+1} = \mathcal{P}_{\mathbb{B}(\mathbf{0}, \epsilon)}\big[\boldsymbol{\delta}(t_k) - \eta \nabla \Phi(\boldsymbol{\delta}(t_k))\big] = \boldsymbol{\delta}(t_k) + \eta \cdot \frac{\epsilon}{\|\boldsymbol{\delta}(t_k) - \eta \nabla \Phi(\boldsymbol{\delta}(t_k))\|}\bigg[ -\nabla \Phi(\boldsymbol{\delta}(t_k)) +$$

$$\boldsymbol{\delta}(t_k) \cdot \langle 2\boldsymbol{\delta}(t_k) - \eta \nabla \Phi(\boldsymbol{\delta}(t_k)), \nabla \Phi(\boldsymbol{\delta}(t_k)) \rangle / \big(\epsilon(\|\boldsymbol{\delta}(t_k)\| + \|\boldsymbol{\delta}(t_k) - \eta \nabla \Phi(\boldsymbol{\delta}(t_k))\|)\big)\bigg], \quad (7)$$

which is consistent with the structure of the explicit single-step method (5). The detailed derivation of formulation (7) is provided in Appendix D.1. Apply Taylor expansion for $\boldsymbol{\delta}(t_{k+1})$, we have:

$$\frac{\boldsymbol{\delta}_{k+1} - \boldsymbol{\delta}_k}{\eta} \approx \frac{\boldsymbol{\delta}(t_k + \eta) - \boldsymbol{\delta}(t_k)}{\eta} = \frac{\partial \boldsymbol{\delta}(t)}{\partial t}\bigg|_{t=t_k} + I_d O(\eta). \quad (8)$$

Combining formulations (7) and (8), $\lim_{\eta \to 0} \big(\frac{\partial \boldsymbol{\delta}(t)}{\partial t}\big|_{t=t_k} + I_d O(\eta)\big)$ is equal to

$$\lim_{\eta \to 0} \frac{\epsilon}{\|\boldsymbol{\delta}(t_k) - \eta \nabla \Phi(\boldsymbol{\delta}(t_k))\|} \cdot \bigg[\boldsymbol{\delta}(t_k) \frac{\langle 2\boldsymbol{\delta}(t_k) - \eta \nabla \Phi(\boldsymbol{\delta}(t_k)), \nabla \Phi(\boldsymbol{\delta}(t_k)) \rangle}{\epsilon(\|\boldsymbol{\delta}(t_k)\| + \|\boldsymbol{\delta}(t_k) - \eta \nabla \Phi(\boldsymbol{\delta}(t_k))\|)} - \nabla \Phi(\boldsymbol{\delta}(t_k))\bigg],$$

$$\Rightarrow \frac{\partial \boldsymbol{\delta}(t)}{\partial t}\bigg|_{t=t_k} = \frac{1}{\epsilon^2}\boldsymbol{\delta}_k \cdot \langle \boldsymbol{\delta}_k, \nabla \Phi(\boldsymbol{\delta}_k) \rangle - \nabla \Phi(\boldsymbol{\delta}_k).$$

Then, we attain ODE (3): $\frac{\partial \boldsymbol{\delta}(t)}{\partial t} = \frac{1}{\epsilon^2}\boldsymbol{\delta}\langle \boldsymbol{\delta}, \nabla \Phi(\boldsymbol{\delta}) \rangle - \nabla \Phi(\boldsymbol{\delta})$.

## 4.2 THE CORRECTNESS OF THE ODE

Since ODE (3) is derived under the conditions of Theorem 3.1, and the trajectory of $\boldsymbol{\delta}_k$ generated by PGD (2) lies on $\mathbb{S}(\mathbf{0}, \epsilon)$, the trajectory of the solution $\boldsymbol{\delta}(t)$ to ODE (3) must also be constrained to $\mathbb{S}(\mathbf{0}, \epsilon)$. Furthermore, the solution $\boldsymbol{\delta}(t)$ to ODE (3) must satisfy the conditions of existence and uniqueness. Additionally, since the derivation in Section 4.1 is informal, we rigorously prove that PGD (2) converges to ODE (3). Therefore, the correctness of ODE (3) is established by addressing the $\mathbb{S}(\mathbf{0}, \epsilon)$ constraint of trajectory, the existence and uniqueness of the solution, and the rigorous convergence from PGD to ODE.

The next theorem ensures that the trajectory of the solution to ODE (3) satisfies $\boldsymbol{\delta}(t) \subset \mathbb{S}(\mathbf{0}, \epsilon), t \geq 0$.

**Theorem 4.1.** *Suppose that (i) $\Phi(\boldsymbol{\delta}) \in \mathcal{F}_M$ and $\boldsymbol{\delta}^* \notin \mathbb{B}(\mathbf{0}, \epsilon)$, (ii) $\epsilon < \frac{\|\nabla \Phi(\mathbf{0})\|}{(U+1)M}$ with $U \geq 2$, (iii) $\boldsymbol{\delta}(0) = \boldsymbol{\delta}_0 \in \mathbb{S}(\mathbf{0}, \epsilon) \cap \mathbb{B}(\boldsymbol{\delta}^\triangle, \sqrt{2}\epsilon)$, we have $\boldsymbol{\delta}(t) \in \mathbb{S}(\mathbf{0}, \epsilon)$, where $\boldsymbol{\delta}(t)$ is the solution to ODE (3).*

See proof in Appendix D.2

Theorem 4.1 aligns with Theorem 3.1, ensuring that the trajectory of $\boldsymbol{\delta}(t)$ satisfies the $\mathbb{S}(\mathbf{0}, \epsilon)$ constraint. Furthermore, Theorem 4.1 guarantees that the assumption in Lemma D.1, which states that every solution of the ODE lies entirely within $W$, is satisfied. Consequently, the existence and uniqueness of the solution to ODE (3) are established based on Lemma D.1 and Theorem 4.1.

**Theorem 4.2** (Existence and Uniqueness). *Suppose that (i) $\Phi(\boldsymbol{\delta}) \in \mathcal{F}_M$ and $\boldsymbol{\delta}^* \notin \mathbb{B}(\mathbf{0}, \epsilon)$, (ii) $\epsilon < \frac{\|\nabla \Phi(\mathbf{0})\|}{(U+1)M}$ with $U \geq 2$, then the ODE (3) with initial condition $\boldsymbol{\delta}_0 \in \mathbb{S}(\mathbf{0}, \epsilon) \cap \mathbb{B}(\boldsymbol{\delta}^\triangle, \sqrt{2}\epsilon)$ has a unique solution $\boldsymbol{\delta}(t) \in C^1([0, \infty); \mathbb{R}^d)$.*

See proof in Appendix D.3

To prove that PGD (2) converges to ODE (3), we introduce the convergence Lemma D.2 in Appendix D.4, which ensures the validity of the ODE derived from the discrete form (5). Using Lemma D.2 and Definition 2.2, we obtain convergence theorem:

**Theorem 4.3.** *Suppose that (i) $\Phi(\boldsymbol{\delta}) \in \mathcal{F}_M$ and $\boldsymbol{\delta}^* \notin \mathbb{B}(\mathbf{0}, \epsilon)$, (ii) $\epsilon < \frac{\|\nabla\Phi(\mathbf{0})\|}{(U+1)M}$ with $U \geq 2$, (iii) $\boldsymbol{\delta}_0 \in \mathbb{S}(\mathbf{0}, \epsilon) \cap \mathbb{B}(\boldsymbol{\delta}^\triangle, \sqrt{2}\epsilon)$, as the step size $\eta \to 0$, PGD (2) converges to ODE (3) in the sense that, for all fixed $T > 0$,*

$$\lim_{\eta \to 0} \max_{0 \leq k \leq \frac{T}{\eta}} \|\boldsymbol{\delta}_k - \boldsymbol{\delta}(k\eta)\| = 0 \tag{9}$$

See proof in Appendix D.5.

Theorem 4.3 rigorously establish the equivalence between PGD (2) and ODE (3), confirming the validity of this transition. Above all, Theorems 4.1, 4.2 and 4.3 collectively substantiate the correctness of ODE (3).

### 4.3 BOUNDS IN CONTINUOUS SCENARIO

To establish the local linear convergence of PGD from a continuous-time perspective, Appendix D.6 introduces Lemma D.3, which is used to demonstrate that the optimal point $\boldsymbol{\delta}^\triangle$ is an exponentially stable equilibrium point of ODE (3). Building on Lemma D.3 and the definition of an exponentially stable equilibrium point, the upper bound for the convergence rate of obtaining $\boldsymbol{\delta}^\triangle$ via ODE (3) is provided below.

**Theorem 4.4.** *Suppose that (i) $\Phi(\boldsymbol{\delta}) \in \mathcal{F}_M$ and $\boldsymbol{\delta}^* \notin \mathbb{B}(\mathbf{0}, \epsilon)$, (ii) and $\epsilon < \frac{\|\nabla\Phi(\mathbf{0})\|}{(U+1)M}$ with $U \geq 2$, (iii) $\boldsymbol{\delta}(0) = \boldsymbol{\delta}_0 \in \mathbb{S}(\mathbf{0}, \epsilon) \cap \mathbb{B}(\boldsymbol{\delta}^\triangle, \sqrt{2}\epsilon)$, if $\Phi(\boldsymbol{\delta})$ is twice continuously differentiable, we have that $\boldsymbol{\delta} = \boldsymbol{\delta}^\triangle$ is an exponentially stable equilibrium point of ODE (3). That is, there exist $r > 0$, $\beta > 0$, $C_4 < 0$ such that if $\|\boldsymbol{\delta}_0 - \boldsymbol{\delta}^\triangle\| < r$, the convergence rate of $\boldsymbol{\delta}(t)$ for finding $\boldsymbol{\delta}^\triangle$ is:*

$$\|\boldsymbol{\delta}(t) - \boldsymbol{\delta}^\triangle\| \leq \beta e^{C_4 t} \|\boldsymbol{\delta}(0) - \boldsymbol{\delta}^\triangle\|, \tag{10}$$

*where $\boldsymbol{\delta}(t)$ is the solution to ODE (3) and $t \geq 0$.*

See proof in Appendix D.6

*Remark* 4.5. Theorem 4.4 implies that when $\boldsymbol{\delta}^* \notin \mathbb{B}(\mathbf{0}, \epsilon)$, the trajectory $\boldsymbol{\delta}(t)$ governed by ODE (3) converges exponentially to $\boldsymbol{\delta}^\triangle$, signifying that the convergence rate of PGD is local linear. Thus, Theorem 4.4 conforms to Theorem 3.1 from the discrete-time scenario, where the initialization can be unified by intersection. Furthermore, according to the analysis in Appendix C.4, this local linear convergence also holds in a general neighborhood, even the initialization is relaxed to $\boldsymbol{\delta}_0 \in \mathbb{B}(\mathbf{0}, \epsilon)$.

## 5 CONNECTIONS AND INTERPRETATIONS

In the discrete-time scenario, PGD (2) is expressed as:

$$\boldsymbol{\delta}_{k+1} = \mathcal{P}_{\mathbb{B}(\mathbf{0}, \epsilon)}[\boldsymbol{\delta}_k - \eta\nabla\Phi(\boldsymbol{\delta}_k)], \ k \geq 0,$$

As outlined in Section 4.1, the connection between ODE and PGD is established. More precisely, PGD (2) in the continuous-time scenario corresponds to the ODE (3):

$$\frac{\partial\boldsymbol{\delta}(t)}{\partial t} = \frac{1}{\epsilon^2}\boldsymbol{\delta}\langle\boldsymbol{\delta}, \nabla\Phi(\boldsymbol{\delta})\rangle - \nabla\Phi(\boldsymbol{\delta}), \ t \geq 0.$$

The existence and uniqueness of the solution $\boldsymbol{\delta}(t)$ to ODE (3) is guaranteed by Theorem 4.2. Additionally, Theorem 4.3 ensures that the trajectory generated by PGD (2) converges rigorously to a curve modeled by ODE (3) as the time step size $\eta \to 0$. The succinct and well-established theory of ODEs provides useful tools for PGD and leads to significant findings, particularly regarding the convergence rate bound in Theorem 4.4. When $\boldsymbol{\delta}^* \notin \mathbb{B}(\mathbf{0}, \epsilon)$, convergence bounds for PGD are derived from both discrete-time and continuous-time perspectives.

In the discrete-time scenario, Theorem 3.1 provides the convergence bounds (6):

$$C_3^k\|\boldsymbol{\delta}_0 - \boldsymbol{\delta}^\triangle\| \leq \|\boldsymbol{\delta}_k - \boldsymbol{\delta}^\triangle\| \leq C_2^k\|\boldsymbol{\delta}_0 - \boldsymbol{\delta}^\triangle\|.$$

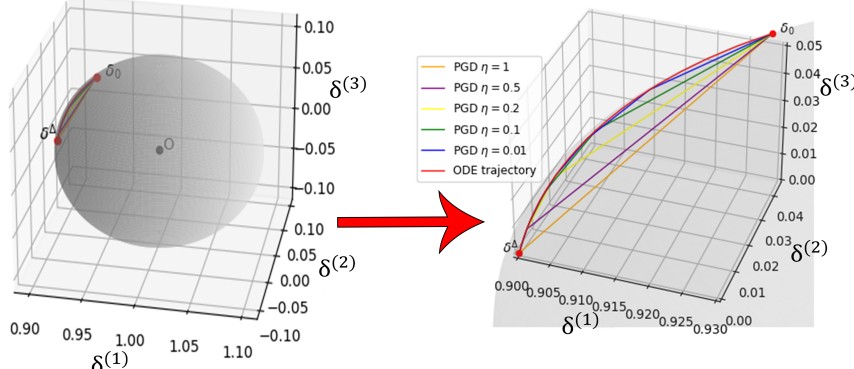

Figure 3: Trajectories determined by PGD (2) and ODE (3) for solving OP (11). In the left figure, the grey ball represents $\mathbb{B}\big((1,0,0)^\top, 0.1\big)$, and the different color curves from $\boldsymbol{\delta}_0$ to $\boldsymbol{\delta}^\triangle$ are the trajectories generated by ODE and PGD. For clarity, the right figure provides an enlarged view of the region where the trajectories are situated.

In the continuous-time scenario, Theorem 4.4 provides the convergence bound (10):

$$\|\boldsymbol{\delta}(t) - \boldsymbol{\delta}^\triangle\| \le \beta e^{C_4 t}\|\boldsymbol{\delta}(0) - \boldsymbol{\delta}^\triangle\|.$$

Since $C_2, C_3 \in (0,1)$ and $C_4 < 0$, the bounds (6) for the discrete-time scenario and the bound (10) for the continuous-time scenario support each other, and consistently demonstrate that if the objective function is convex and smooth, the convergence rate of PGD for solving OP (1) is local linear. Equivalently, Theorems 3.1 and 4.4 corroborate one another.

## 6    EXPERIMENTS

In this section, we conduct numerical experiments to validate our theoretical findings. Specifically, in Section 6.1, we verify that the limit of PGD (2) corresponds to ODE (3), which aligns with Theorem 4.3. In Section 6.2, we validate the local linear convergence of PGD from discrete-time and continuous-time scenarios. Additional experiments on linear, log-sum-exp, or logistic regression loss functions are presented in Appendix E. The observed experimental phenomenon of PGD's local linear convergence corroborates Theorems 3.1 and 4.4. All simulations are implemented on a PC with an Intel(R) Core(TM) i7-9700 CPU, taking less than 1 minute.

### 6.1    ODE VERIFICATION

To substantiate the equivalence between PGD (2) and ODE (3), we examine the following OP, leveraging the translation invariance of functions for simplicity:

$$\min_{\boldsymbol{\delta}\in\mathbb{B}\big((1,0,0)^\top,0.1\big)} \boldsymbol{\delta}^\top\boldsymbol{\delta}/2, \tag{11}$$

where $\boldsymbol{\delta} = (\delta^{(1)}, \delta^{(2)}, \delta^{(3)})^\top$. For OP (11), we have $\boldsymbol{\delta}^\triangle = (0.9, 0, 0)^\top$ and $\boldsymbol{\delta}^* = (0, 0, 0)^\top \notin \mathbb{B}((1,0,0)^\top, 0.1)$. Based on ODE (3), we obtain the following ODE that corresponds to PGD (2) for solving OP (11):

$$\partial\boldsymbol{\delta}(t)/\partial t = -100(\boldsymbol{\delta} - (1,0,0)^\top)\langle\boldsymbol{\delta} - (1,0,0)^\top, \boldsymbol{\delta}\rangle + \boldsymbol{\delta}. \tag{12}$$

For PGD, we set $\eta \in \{1, 0.5, 0.2, 0.1, 0.01\}$. For ODE (12), we set the terminal evolutionary time $T = 100$ and $t \in [0, T]$. Both PGD (2) and ODE (12) share a common initial point $\boldsymbol{\delta}_0$. Because any $\boldsymbol{\delta}_0$ on the surface of the constraint ball and in proximity to $\boldsymbol{\delta}^\triangle$ is considered feasible, we use the following value: $\boldsymbol{\delta}_0 = \mathcal{P}_{\mathbb{B}\big((1,0,0)^\top,0.1\big)}\big[(1 + \cos\frac{5\pi}{6}, \sin\frac{5\pi}{6}, \sin\frac{3\pi}{4})^\top\big]$.

For OP (11), the trajectories of $\boldsymbol{\delta}_k$ generated by PGD with various $\eta$ and the trajectory of $\boldsymbol{\delta}(t)$ determined by ODE (12) are shown in Figure 3. As observed in the red curve shown in Figure 3, the trajectory of $\boldsymbol{\delta}(t)$ determined by ODE (12) is on $\mathbb{S}\big((1,0,0)^\top, 0.1\big)$, which validate our Theorem 4.1. Besides, Figure 3 indicates that as $\eta$ decreases, the trajectories of $\boldsymbol{\delta}_k$ generated by PGD converge to the trajectory of $\boldsymbol{\delta}(t)$ determined by ODE, confirming our Theorem 4.3. Based on the analysis of Figure 3, ODE (3) exhibits approximate equivalence to the PGD method (2) and thereby enhances our comprehension of PGD from a continuous-time scenario.

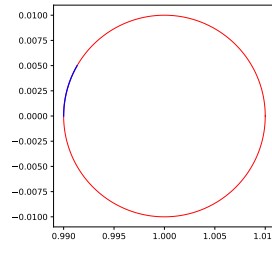

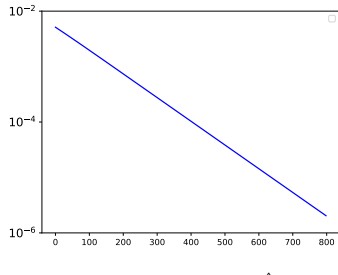

(a) Trajectory of $\boldsymbol{\delta}_k$ (blue line)

(b) Error: $\|\boldsymbol{\delta}_k - \boldsymbol{\delta}^\triangle\|$

Figure 4: Trajectory and Error of $\boldsymbol{\delta}_k$ produced by PGD for $\epsilon = 0.01$. The vertical axis in Figure (b) represents $\log_{10}(\|\boldsymbol{\delta}_k - \boldsymbol{\delta}^\triangle\|)$.

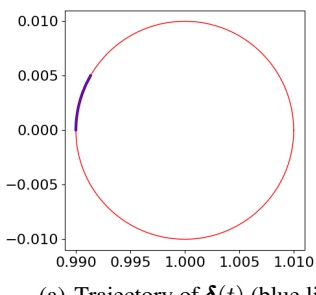

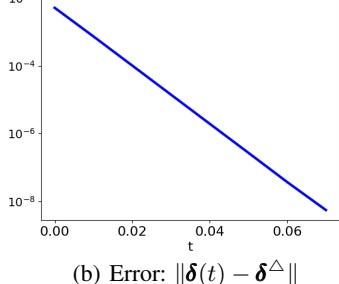

(a) Trajectory of $\boldsymbol{\delta}(t)$ (blue line)

(b) Error: $\|\boldsymbol{\delta}(t) - \boldsymbol{\delta}^\triangle\|$

Figure 5: Trajectory and Error of $\boldsymbol{\delta}(t)$ produced by ODE for $\epsilon = 0.01$. The vertical axis Figure (b) represents $\log_{10}(\|\boldsymbol{\delta}(t) - \boldsymbol{\delta}^\triangle\|)$.

## 6.2 Linear Convergence of PGD

### 6.2.1 Fixed Constraint Radius

In this chapter, we verify that the convergence rate of PGD is linear in both discrete-time and continuous-time scenarios with the objective function convex and smooth. To simplify our analysis, we consider the following OP, taking into account the translation invariance of functions:

$$\min_{\boldsymbol{\delta} \in \mathbb{B}((1,0)^\top, \epsilon)} \Phi(\boldsymbol{\delta}), \tag{13}$$

where $\Phi(\boldsymbol{\delta}) = \boldsymbol{\delta}^\top \begin{pmatrix} 1 & 0 \\ 0 & 0 \end{pmatrix} \boldsymbol{\delta}$ and $\boldsymbol{\delta} = (\delta^{(1)}, \delta^{(2)})^\top$. Because, $\nabla^2 \Phi(\boldsymbol{\delta}) = \begin{pmatrix} 2 & 0 \\ 0 & 0 \end{pmatrix}$, both $\nabla^2 \Phi(\boldsymbol{\delta})$ and $2E - \nabla^2 \Phi(\boldsymbol{\delta})$ are positive semidefinite. Therefore, $\Phi(\boldsymbol{\delta})$ is 2-smooth and convex. In light of the conditions in Theorems 3.1 and 4.4, $\epsilon < \frac{\|\nabla \Phi((1,0)^\top)\|}{3M} = \frac{\|(2,0)^\top\|}{3 \times 2} = \frac{1}{3}$, $\eta < \frac{1}{2M} = \frac{1}{4}$. Thus, we set the constraint radius $\epsilon \leq 0.2$, and the step size $\eta = 5 \times 10^{-5}$. Then, we have $\boldsymbol{\delta}^\triangle = (1 - \epsilon, 0)^\top$ and $\boldsymbol{\delta}^* = (0, c)^\top \notin \mathbb{B}((1,0)^\top, \epsilon)$, where $c \in \mathbb{R}$. Moreover, $\boldsymbol{\delta}_0$ is initialized as $(1 - \frac{\sqrt{2}}{2}\epsilon, \frac{\sqrt{2}}{2}\epsilon)^\top$ for convenience. Under these settings, using PGD to solve (13) satisfies the conditions of Theorems 3.1 and 4.4. For convenience, we set $\epsilon = 0.01$ in OP (13). Then, the optimal point of OP (13) is $\boldsymbol{\delta}^\triangle = (0.99, 0)^\top$ and $\boldsymbol{\delta}_0 = (1 - \frac{\sqrt{2}}{200}, \frac{\sqrt{2}}{200})^\top$. Next, we verify the linear convergence of PGD from the discrete-time and continuous-time perspectives.

From the perspective of the discrete-time scenario, we directly apply PGD (2) with $\eta = 5 \times 10^{-5}$ to solve OP (13). The trajectory and error of $\boldsymbol{\delta}_k$ generated by PGD are displayed in Figure 4. As depicted in Figure 4 (a), the blue trajectory of $\boldsymbol{\delta}_k$ is confined to $\mathbb{S}((1,0)^\top, 0.01)$ and converges to $(0.99, 0)^\top$, which is consistent with Lemma C.1. Additionally, the plot in Figure 4 (b) is a straight line, indicating that $\boldsymbol{\delta}_k$ generated by PGD converges to $\boldsymbol{\delta}^\triangle$ at a linear rate. These results in Figure 4 are in perfect agreement with Theorems 3.1.

From the perspective of the continuous-time scenario motivated by ODE (3), we obtain the following ODE that corresponds to PGD (2) for solving OP (13):

$$\partial \boldsymbol{\delta}(t)/\partial t = -2 \times 10^4 \cdot \delta^{(1)}(\delta^{(1)} - 1)(\delta^{(1)} - 1, \delta^{(2)})^\top + (2\delta^{(1)}, 0)^\top. \tag{14}$$

The trajectory and error of $\boldsymbol{\delta}(t)$ determined by ODE (14) are depicted in Figure 5. As illustrated in Figure 5 (a), the blue trajectory of $\boldsymbol{\delta}(t)$ is on $\mathbb{S}((1,0)^\top, 0.01)$ and converges to $(0.99, 0)^\top$, which

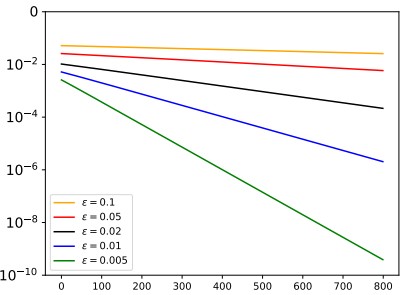 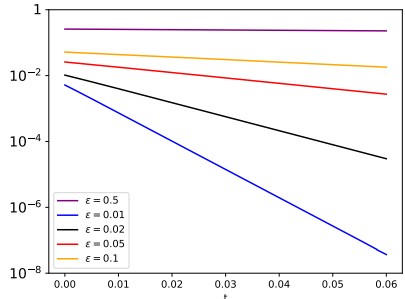

Figure 6: The convergence of $\boldsymbol{\delta}_k$ produced by PGD for solving OP (13) with different $\epsilon$. The horizontal axis represents the number of iterations $k$ and the vertical axis represents $\log_{10}(\|\boldsymbol{\delta}_k - \boldsymbol{\delta}^\triangle\|)$.

Figure 7: The convergence of $\boldsymbol{\delta}(t)$ decided by ODE (3) under different $\epsilon$ for solving OP (13). The horizontal axis represents evolutionary time $t$ and the vertical axis represents $\log_{10}(\|\boldsymbol{\delta}(t) - \boldsymbol{\delta}^\triangle\|)$.

aligns with the results in Theorem 4.1. Besides, the plot in Figure 5 (b) is a straight line, indicating that $\boldsymbol{\delta}(t)$ generated by ODE (14) converges to $\boldsymbol{\delta}^\triangle$ at a linear rate. These results in Figure 5 are in perfect accordance with Theorem 4.4.

It is worth noting that although Figures 4 and 5 appear similar, they are fundamentally distinct: Figure 4 corresponds to $\boldsymbol{\delta}_k$ generated by PGD in the discrete-time scenario, while Figure 5 corresponds to $\boldsymbol{\delta}(t)$ determined by the ODE in the continuous-time scenario. Additionally, the horizontal axis in Figure 4 (b) represents iterations $k$, whereas in Figure 5 (b), the horizontal axis is time $t$, where $k \leq 800$ and $t \leq 0.8$.

Comparing Figure 4 (a) with Figure 5 (a), the trajectories of $\boldsymbol{\delta}_k$ and $\boldsymbol{\delta}(t)$ coincide, indicating that ODE (3) exhibits approximate equivalence to PGD method (2) when the step size $\eta$ is sufficiently small. This observation verifies our Theorem 4.3. Moreover, comparing Figure 4 (b) with Figure 5 (b), they are both straight lines, which means that the convergence rate of PGD is linear, irrespective of whether considered from a discrete-time or continuous-time perspective. This illustrates that Theorem 4.4 derived from the continuous-time scenario is consistent with Theorem 3.1 derived from the discrete-time scenario.

### 6.2.2 DIFFERENT CONSTRAINT RADIUS

To further validate the linear convergence rate of PGD, we apply PGD method (2) and ODE (3) to solve OP (13) under $\epsilon \in \{0.5, 0.1, 0.05, 0.02, 0.01, 0.005\}$ with $\eta = 5 \times 10^{-5}$. The convergent curves obtained by PGD under various $\epsilon$ are depicted in Figure 6. The convergent curves determined by ODE (3) under various $\epsilon$ are displayed in Figure 7. The observations from both Figures 6 and 7 are as follows: The plots for different $\epsilon$ are all straight lines, implying that PGD exhabits linear convergence rate in both the discrete-time and continuous-time scenarios. Combining the analysis from both Figures 6 and 7, we conclude that Theorem 4.4 from the continuous-time scenario and Theorem 3.1 from the discrete-time scenario support each other. Together, they consistently demonstrate that PGD exhibits linear convergence when the objective function is convex and smooth. Additional experiments presented in Appendix E confirm the same conclusions as those from Figures 6 and 7.

## 7 CONCLUSION

In this paper, we establish an ODE (3) as the continuous-time counterpart of PGD (2). The correctness of ODE (3) is rigorously verified through the $\mathbb{S}(\mathbf{0}, \epsilon)$ constraint of trajectory, the existence and uniqueness of the solution, and the convergence validity from PGD to ODE. (i.e., Theorems 4.1, 4.2 and 4.3). Additionally, we provide the convergence rate bounds for PGD in both discrete-time and continuous-time scenarios, where the latter is inspired by the ODE framework (i.e., Theorems 3.1 and 4.4). The convergence rate bounds from both scenarios mutually reinforce each other and consistently affirm that: when the objective function $\Phi(\boldsymbol{\delta})$ is convex and smooth and $\boldsymbol{\delta}^* \notin \mathbb{B}(\mathbf{0}, \epsilon)$, the convergence rate of PGD (2) for solving OP(1) is local linear. Our experimental results provide empirical support for these theoretical findings.

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

# Appendix

## A  RELATED WORK

### A.1  OPTIMIZATION METHODS

Optimization significantly promotes the development of many fields in machine learning, such as reinforcement learning (Kim et al., 2024), generative networks (Robey et al., 2021), and adversarial attacks (Cheng et al., 2024), etc. Many algorithms and techniques have been introduced to address optimization problems, such as Newton's methods (Smee & Roosta, 2024), trust-region methods (Song et al., 2023), interior point methods (Chowdhury et al., 2020), and PGD methods (Fan et al., 2024). Among these, the PGD method stands out as a valuable algorithm for constrained optimization (Boyd et al., 2011; Beck, 2014; Jain & Kar, 2017), and has demonstrated substantial success in practical applications (Muehlebach & Jordan, 2022; Diao et al., 2025; Li et al., 2024).

The basic idea of PGD is extensively covered in textbooks (Nesterov, 2018; Nocedal & J. Wright, 2006; Jain & Kar, 2017). From a theoretical perspective, the analysis of PGD closely resembles that of gradient descent methods, with a generalization of the gradient concept into "gradient mapping" (Nesterov, 2004). In addition, due to their shared objective of finding the optimal point, PGD method can be replaced with the Lagrange multiplier method (Haeser et al., 2021), Frank-Wolfe algorithm (Gonçalves et al., 2024), and sequential quadratic programming (Torrisi et al., 2018; Rashidi & Soleimani-Damaneh, 2024). It is worth emphasizing that the PGD method differs from the Lagrange multiplier method, although they can replace each other to solve the same constrained optimization (Vu & Raich, 2022). In terms of convergence research, the paper (Nesterov, 2018) demonstrates that the convergence rate of PGD under simple sets is sublinear for general convex and smooth objective functions. These simple sets encompass positive orthants, $d$-dimensional boxes, simplexes, and Euclidean balls.

### A.2  ODE AND OPTIMIZATION

In a different direction, there is a rich history relating ordinary differential equations to optimization, (Wang & Wu, 2020; Chen et al., 2018; Song & Khan, 2022). The connection between ODEs and numerical optimization is often established by taking step sizes to be very small, causing the trajectory or solution path to converge to a curve modeled by an ODE (Arnaboldi et al., 2023). The concise and well-established theory of ODEs offers profound views into the optimization field and has led to many intriguing discoveries. Some noteworthy examples include linear regression via solving differential equations induced by linearized Bregman iteration algorithm (Osher et al., 2016), a continuous-time Nesterov-like algorithm in the context of control design (Dürr & Ebenbauer, 2012; Durr et al., 2012), and modeling design iterative optimization algorithms as nonlinear dynamical systems (Lessard et al., 2016). Additionally, paper (Su et al., 2016) proved that Nesterov's accelerated gradient method can be derived from a second-order ODE. Moreover, the work (Su et al., 2016) gets many properties of Nesterov's accelerated gradient method from the perspective of ODE. As a result, ODEs offer a new perspective for the study of the convergence rate of the PGD when applied to solving optimization problems. In this paper, we investigate the PGD method from discrete-time and continuous-time scenarios, where the continuous-time analysis is inspired by the ODE.

## B  LIMITATION

This paper explores the local linear convergence properties of PGD for solving optimization problems with a general convex and smooth objective function constrained by $\mathbb{B}(\mathbf{0}, \epsilon)$ from both discrete and continuous perspectives. For general convex constraints (e.g., polytopes, simplices), PGD typically has sublinear convergence, and establishing linear rates is often infeasible. By focusing on a structured case ($l_2$ ball), our work provides a first-step theoretical foundation for understanding how PGD can exhibit local linear convergence, particularly from an ODE perspective. This convergence behavior is not addressed by existing analyses. We will explore the local linear convergence of PGD for more general constraint settings in future work.

## C PROOFS OF THEOREM 3.1

In this section, we present the proofs of Theorem 3.1. Specifically, Section C.1 provides the necessary preliminaries for the proofs. Subsequently, the proofs for the upper and lower bounds are detailed in Sections C.2 and C.3, respectively.

*Theorem* 3.1 (Upper and Lower Bounds). Suppose that (i) $\Phi(\boldsymbol{\delta}) \in \mathcal{F}_M$ and $\boldsymbol{\delta}^* \notin \mathbb{B}(\mathbf{0}, \epsilon)$, (ii) $\epsilon < \frac{\|\nabla\Phi(\mathbf{0})\|}{(U+1)M}$ with $U \geq 2$, (iii) $\eta < \frac{1}{2M}$, (iv) $\boldsymbol{\delta}_k$ is generated by the PGD (2). Then, there exists $\theta \in (0, \frac{\pi}{3})$, if $\boldsymbol{\delta}_0 \in \mathbb{S}(\mathbf{0}, \epsilon) \cap \mathbb{B}(\boldsymbol{\delta}^\triangle, 2\epsilon \sin\frac{\theta}{2})$, we have $\boldsymbol{\delta}_k \in \mathbb{S}(\mathbf{0}, \epsilon)$ for $k \in \mathbb{Z}^+$ and

$$C_3^k \|\boldsymbol{\delta}_0 - \boldsymbol{\delta}^\triangle\| \leq \|\boldsymbol{\delta}_k - \boldsymbol{\delta}^\triangle\| \leq C_2^k \|\boldsymbol{\delta}_0 - \boldsymbol{\delta}^\triangle\|, \tag{15}$$

where $C_2 = \frac{2+3\eta M}{2+2U\eta M} < 1$ and $C_3 = \frac{2-3\eta M}{2+2U\eta M} < C_2 < 1$.

### C.1 PRELIMINARIES FOR PROOF OF BOUNDS

In this chapter, we introduce the specific process of PGD from a geometric perspective and provide a lemma to guarantee $\boldsymbol{\delta}_k \in \mathbb{S}(\mathbf{0}, \epsilon)$ for $k \in \mathbb{Z}^+$.

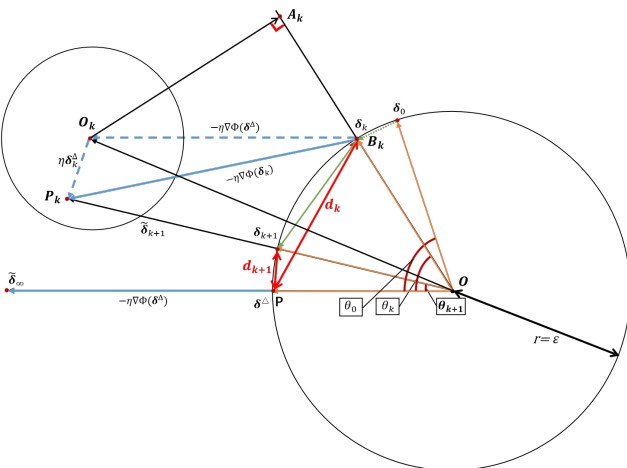

Figure 8: A specific magnified version of $\mathbb{B}(\mathbf{0}, \epsilon)$ in Figure 2 (a) when $\boldsymbol{\delta}^* \notin \mathbb{B}(\mathbf{0}, \epsilon)$. The orange arrow represents $\boldsymbol{\delta}_k$. The green trajectory of $\boldsymbol{\delta}_k$ is generated by PGD. The solid blue arrow means $-\eta\nabla\Phi(\boldsymbol{\delta}_k)$. The dotted blue arrows indicate auxiliary vectors $-\eta\nabla\Phi(\boldsymbol{\delta}^\triangle)$ and $\eta\boldsymbol{\delta}_k^\triangle$, where $\boldsymbol{\delta}_k^\triangle :=$ $\nabla\Phi(\boldsymbol{\delta}^\triangle) - \nabla\Phi(\boldsymbol{\delta}_k)$. The red line segment corresponds to $d_k = \|\boldsymbol{\delta}_k - \boldsymbol{\delta}^\triangle\|$.

Because $\Phi(\boldsymbol{\delta}) \in \mathcal{F}_M$ with $\boldsymbol{\delta}^* \notin \mathbb{B}(\mathbf{0}, \epsilon)$, $\mathbb{B}(\mathbf{0}, \epsilon)$ is externally tangential to some contour line of $\Phi(\boldsymbol{\delta})$ at $\boldsymbol{\delta}^\triangle$. According to the analysis on case $\boldsymbol{\delta}^* \notin \mathbb{B}(\mathbf{0}, \epsilon)$ in Section 2.1, we have $\boldsymbol{\delta}^\triangle \in \mathbb{S}(\mathbf{0}, \epsilon)$ and $-\nabla\Phi(\boldsymbol{\delta}^\triangle) = \lambda\boldsymbol{\delta}^\triangle$ for some $\lambda > 0$. And,

$$\|\nabla\Phi(\boldsymbol{\delta}^\triangle)\| = \lambda\|\boldsymbol{\delta}^\triangle\| = \lambda\epsilon, \tag{16}$$

$$\nabla\Phi(\boldsymbol{\delta}_k) = \nabla\Phi(\boldsymbol{\delta}^\triangle) - \boldsymbol{\delta}_k^\triangle = -\lambda\boldsymbol{\delta}^\triangle - \boldsymbol{\delta}_k^\triangle. \tag{17}$$

Suppose $\boldsymbol{\delta}_k \in \mathbb{S}(\mathbf{0}, \epsilon)$, $\theta_k$ is the angle between $\boldsymbol{\delta}_k$ and $\boldsymbol{\delta}^\triangle$, and $\theta_k \in [0, \frac{\pi}{3}]$. As shown in Figure 8,

$$\overrightarrow{OP} = \boldsymbol{\delta}^\triangle, \overrightarrow{OB_k} = \boldsymbol{\delta}_k, \overrightarrow{B_kO_k} = -\eta\nabla\Phi(\boldsymbol{\delta}^\triangle), \overrightarrow{O_kP_k} = \eta\boldsymbol{\delta}_k^\triangle, \tag{18}$$

$$\overrightarrow{OO_k} = \overrightarrow{OB_k} + \overrightarrow{B_kO_k} = \boldsymbol{\delta}_k - \eta\nabla\Phi(\boldsymbol{\delta}^\triangle) = \boldsymbol{\delta}_k + \lambda\eta\boldsymbol{\delta}^\triangle, \tag{19}$$

$$\overrightarrow{OP_k} = \overrightarrow{OO_k} + \overrightarrow{O_kP_k} = \boldsymbol{\delta}_k + \eta(\lambda\boldsymbol{\delta}^\triangle + \boldsymbol{\delta}_k^\triangle) = \boldsymbol{\delta}_k - \eta\nabla\Phi(\boldsymbol{\delta}_k) = \widetilde{\boldsymbol{\delta}}_{k+1}. \tag{20}$$

According to PGD method (2) and formulation (20), $\theta_{k+1} = \angle POP_k$. Under the assumption $\Phi(\boldsymbol{\delta}) \in \mathcal{F}_M$,

$$
\begin{aligned}
\|\boldsymbol{\delta}_k^{\triangle}\| =& \|\nabla\Phi(\boldsymbol{\delta}_k) - \nabla\Phi(\boldsymbol{\delta}^{\triangle})\| \leq M\|\boldsymbol{\delta}_k - \boldsymbol{\delta}^{\triangle}\| \\
=& M(\langle\boldsymbol{\delta}_k, \boldsymbol{\delta}_k\rangle + \langle\boldsymbol{\delta}^{\triangle}, \boldsymbol{\delta}^{\triangle}\rangle - 2\langle\boldsymbol{\delta}_k, \boldsymbol{\delta}^{\triangle}\rangle)^{\frac{1}{2}} \\
=& 2M\epsilon\sin\frac{\theta_k}{2},
\end{aligned}
\tag{21}
$$

and

$$
\|\nabla\Phi(\mathbf{0}) - \nabla\Phi(\boldsymbol{\delta}^{\triangle})\| \leq M\|\boldsymbol{\delta}^{\triangle} - \mathbf{0}\| \Rightarrow \|\nabla\Phi(\mathbf{0})\| - M\epsilon \leq \|\nabla\Phi(\boldsymbol{\delta}^{\triangle})\|.
\tag{22}
$$

Combine (18), (20) and (21), we have

$$
\|\overrightarrow{O_kP_k}\| = \eta\|\boldsymbol{\delta}_k^{\triangle}\| \leq 2\eta M\epsilon\sin\frac{\theta_k}{2},
$$

$$
\overrightarrow{OP_k} = \widetilde{\boldsymbol{\delta}}_{k+1} \in \mathbb{B}(O_k, 2\eta M\epsilon\sin\frac{\theta_k}{2}).
\tag{23}
$$

Subsequently, we give a lemma to guarantee $\boldsymbol{\delta}_k \in \mathbb{S}(\mathbf{0}, \epsilon)$ for $k \in \mathbb{Z}^+$.

**Lemma C.1.** *Suppose that (i) $\Phi(\boldsymbol{\delta}) \in \mathcal{F}_M$ and $\boldsymbol{\delta}^* \notin \mathbb{B}(\mathbf{0}, \epsilon)$, (ii) $\epsilon < \frac{\|\nabla\Phi(\mathbf{0})\|}{(U+1)M}$ with $U \geq 2$, (iii) $\eta < \frac{1}{2M}$, (iv) $\boldsymbol{\delta}_k$ is generated by PGD (2) for solving OP (1). Then, $\forall\theta \in (0, \frac{\pi}{3})$, if $\boldsymbol{\delta}_k \in \mathbb{S}(\mathbf{0}, \epsilon) \cap \mathbb{B}(\boldsymbol{\delta}^{\triangle}, 2\epsilon\sin\frac{\theta}{2})$, we have $\boldsymbol{\delta}_{k+1} \in \mathbb{S}(\mathbf{0}, \epsilon) \cap \mathbb{B}(\boldsymbol{\delta}^{\triangle}, 2\epsilon\sin\frac{\theta_k}{2})$.*

*Proof.* According to $\epsilon < \frac{\|\nabla\Phi(\mathbf{0})\|}{(U+1)M}$ with $U \geq 2$ in $(ii)$ and formulation (22),

$$
\epsilon < \frac{\|\nabla\Phi(\mathbf{0})\| - M\epsilon}{UM} < \frac{\|\nabla\Phi(\boldsymbol{\delta}^{\triangle})\|}{UM}.
\tag{24}
$$

Combine (16) and (24),

$$
\epsilon = \frac{\|\nabla\Phi(\boldsymbol{\delta}^{\triangle})\|}{\lambda} < \frac{\|\nabla\Phi(\boldsymbol{\delta}^{\triangle})\|}{UM},
$$

and $\lambda > UM$ holds. As shown in Figure 8, for $\triangle O_kA_kB_k$, $O_kA_k \perp B_kA_k$. Based on (16) and (18),

$$
\|\overrightarrow{O_kA_k}\| = \|\overrightarrow{B_kO_k}\|\cos\angle A_kO_kB_k = \epsilon\eta\lambda\sin\theta_k > 2\epsilon\eta M\sin\frac{\theta_k}{2} \geq \|\overrightarrow{O_kP_k}\|.
\tag{25}
$$

(25) guarantees $\gamma\boldsymbol{\delta}_k \cap \mathbb{B}(O_k, 2\eta M\epsilon\sin\frac{\theta_k}{2}) = \varnothing$, where $\gamma \in \mathbb{R}$. Suppose $\alpha_k$ is the angle between $\boldsymbol{\delta}_k^{\triangle}$ and $\boldsymbol{\delta}_k$. Since $\theta_k \in (0, \frac{\pi}{3})$, according to (17),

$$
\begin{aligned}
\langle-\nabla\Phi(\boldsymbol{\delta}_k), \boldsymbol{\delta}_k\rangle =& \langle\boldsymbol{\delta}_k^{\triangle} - \nabla\Phi(\boldsymbol{\delta}^{\triangle}), \boldsymbol{\delta}_k\rangle = \langle\lambda\boldsymbol{\delta}^{\triangle} + \boldsymbol{\delta}_k^{\triangle}, \boldsymbol{\delta}_k\rangle \\
=& \lambda\|\boldsymbol{\delta}^{\triangle}\| \cdot \|\boldsymbol{\delta}_k\|\cos\theta_k + \|\boldsymbol{\delta}_k^{\triangle}\| \cdot \|\boldsymbol{\delta}_k\|\cos\alpha_k \\
\geq& \lambda\epsilon^2\cos\theta_k - 2M\epsilon^2\sin\frac{\theta_k}{2} \\
=& -\epsilon^2(2\lambda\sin^2\frac{\theta_k}{2} + 2M\sin\frac{\theta_k}{2} - \lambda) > 0.
\end{aligned}
\tag{26}
$$

When $\eta$ satisfy condition (iii), based on (23),

$$
\|\overrightarrow{O_kP_k}\| \leq 2\eta M\epsilon\sin\frac{\theta_k}{2} \leq \epsilon\sin\theta_k.
\tag{27}
$$

Therefore, (27) guarantees $\gamma\boldsymbol{\delta}^{\triangle} \cap \mathbb{B}(O_k, 2\eta M\epsilon\sin\frac{\theta_k}{2}) = \varnothing$, where $\gamma \in \mathbb{R}$. Based on (25), (26), and (27),

$$
\mathbb{B}(O_k, 2\eta M\epsilon\sin\frac{\theta_k}{2}) \subset \{\widetilde{\boldsymbol{\delta}}|0 \leq \angle(\widetilde{\boldsymbol{\delta}}, \boldsymbol{\delta}^{\triangle}) < \theta_k, \widetilde{\boldsymbol{\delta}} \notin \mathbb{B}(\mathbf{0}, \epsilon)\},
$$

where $\angle(\widetilde{\boldsymbol{\delta}}, \boldsymbol{\delta}^{\triangle})$ denotes the angle between $\widetilde{\boldsymbol{\delta}}$ and $\boldsymbol{\delta}^{\triangle}$. Therefore, according to (2) and (23), PGD project $\widetilde{\boldsymbol{\delta}}_{k+1}$ to $\boldsymbol{\delta}_{k+1}$ on $\mathbb{S}(\mathbf{0}, \epsilon)$, and we have $\boldsymbol{\delta}_{k+1} \in \mathbb{S}(\mathbf{0}, \epsilon) \cap \mathbb{B}(\boldsymbol{\delta}^{\triangle}, 2\epsilon\sin\frac{\theta_k}{2})$.

$\square$

Next, we prove the upper and lower bounds in Theorems 3.1 by finding the minimum and maximum of $\theta_{k+1}$ when $\boldsymbol{\delta}_k$ is fixed.

## C.2 Proof of Upper Bound

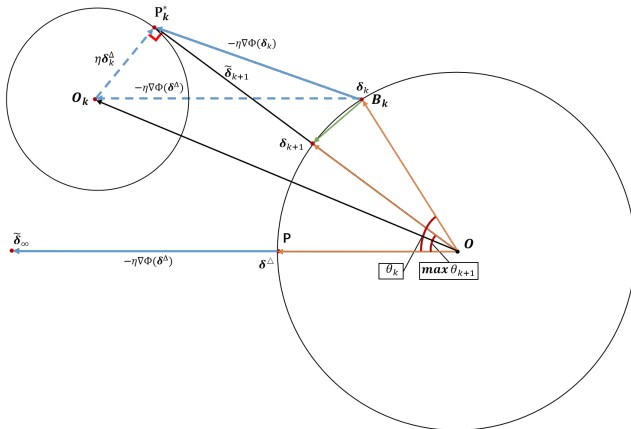

Figure 9: The situation that $\theta_{k+1}$ gets maximum in Figure 8. In this situation, $OP_k$ is externally tangential to $\mathbb{B}(O_k, 2\eta M\epsilon \sin \frac{\theta_k}{2})$ at $P_k^*$. Meanwhile, lines $OP, OO_k$, and $OP_k^*$ are on the same plane.

In this chapter, we present the proof of the upper bound for the convergence rate of PGD. Based on formulation (23) and Figure 8,

$$\theta_{k+1} = \angle P_k OP \leq \angle P_k OO_k + \angle O_k OP, \tag{28}$$

where $O$, $O_k$ and $P$ are fixed point; $P_k \in \mathbb{B}(O_k, 2\eta M\epsilon \sin \frac{\theta_k}{2})$. Thus, $\angle O_k OP$ is fixed and $\max \theta_{k+1} = \angle O_k OP + \max \angle P_k OO_k$. $\angle O_k OP_k$ gets maximum when $OP_k$ is externally tangential to $\mathbb{B}(O_k, 2\eta M\epsilon \sin \frac{\theta_k}{2})$. Thus, the situation that $\theta_{k+1}$ gets maximum is shown in Figure 9 where $P_k = P_k^*$. As illustrated in Figure 9, $\overrightarrow{OP_k^*} \perp \overrightarrow{O_k P_k^*}$ with $\|\overrightarrow{O_k P_k^*}\| = 2\eta M\epsilon \sin \frac{\theta_k}{2}$. According to (19),

$$\|\overrightarrow{OO_k}\| = \sqrt{\langle \boldsymbol{\delta}_k + \lambda\eta\boldsymbol{\delta}^\triangle, \boldsymbol{\delta}_k + \lambda\eta\boldsymbol{\delta}^\triangle \rangle} = \epsilon(1 + \lambda^2\eta^2 + 2\lambda\eta \cos\theta_k)^{\frac{1}{2}}. \tag{29}$$

As shown in Figure 9, $\angle OP_k^* O_k = \frac{\pi}{2}$ in $\triangle O_k P_k^* O$. Therefore, based on (18), (21) and (29),

$$\sin \angle O_k OP_k^* = \|\overrightarrow{O_k P_k^*}\| \big/ \|\overrightarrow{OO_k}\| = \frac{2\eta M \sin \frac{\theta_k}{2}}{(1 + \lambda^2\eta^2 + 2\lambda\eta \cos\theta_k)^{\frac{1}{2}}}, \tag{30}$$

$$\cos \angle O_k OP_k^* = \sqrt{1 - \cos^2 \angle O_k OP_k^*} = \left( \frac{1 + \lambda^2\eta^2 + 2\lambda\eta \cos\theta_k - 4\eta^2 M^2 \sin^2 \frac{\theta_k}{2}}{1 + \lambda^2\eta^2 + 2\lambda\eta \cos\theta_k} \right)^{\frac{1}{2}}. \tag{31}$$

Besides, based on (18), (19) and (29),

$$\cos \angle O_k OP = \frac{\langle \overrightarrow{OO_k}, \overrightarrow{OP} \rangle}{\|\overrightarrow{OO_k}\| \cdot \|\overrightarrow{OP}\|} = \frac{\langle \boldsymbol{\delta}_k + \lambda\eta\boldsymbol{\delta}^\triangle, \boldsymbol{\delta}^\triangle \rangle}{\|\overrightarrow{OO_k}\| \cdot \|\boldsymbol{\delta}^\triangle\|} = \frac{\eta\lambda + \cos\theta_k}{(1 + \lambda^2\eta^2 + 2\lambda\eta \cos\theta_k)^{\frac{1}{2}}}, \tag{32}$$

$$\sin \angle O_k OP = \sqrt{1 - \sin^2 \angle O_k OP} = \frac{\sin\theta_k}{(1 + \lambda^2\eta^2 + 2\lambda\eta \cos\theta_k)^{\frac{1}{2}}}. \tag{33}$$

From the analysis on (25) and (28),

$$\theta_{k+1} \leq \angle P_k^* OO_k + \angle O_k OP \leq \frac{\pi}{3}. \tag{34}$$

Combine (30) $\sim$ (34),

$$\begin{aligned}
\cos\theta_{k+1} &\geq \cos\left( \angle P_k^* OO_k + \angle O_k OP \right) \\
&= \cos \angle P_k^* OO_k \cos \angle O_k OP - \sin \angle P_k^* OO_k \sin \angle O_k OP \\
&= \left[ (1 + \eta^2\lambda^2 + 2\eta\lambda \cos\theta_k - 4\eta^2 M^2 \sin^2 \frac{\theta_k}{2})^{\frac{1}{2}} (\eta\lambda + \cos\theta_k) \right. \\
&\quad \left. - 2\eta M \sin \frac{\theta_k}{2} \sin\theta_k \right] \Big/ (1 + \eta^2\lambda^2 + 2\eta\lambda \cos\theta_k)
\end{aligned} \tag{35}$$

As the red line segment shown in Figure 8,

$$d_k = \|\boldsymbol{\delta}_k - \boldsymbol{\delta}^\triangle\| = \langle \boldsymbol{\delta}_k - \boldsymbol{\delta}^\triangle, \boldsymbol{\delta}_k - \boldsymbol{\delta}^\triangle \rangle^{\frac{1}{2}} = 2\epsilon \sin \frac{\theta_k}{2}. \tag{36}$$

Based on (35) and (36), we have

$$\frac{d_{k+1}}{d_k} = 2\epsilon \sin \frac{\theta_{k+1}}{2} \Big/ (2\epsilon \sin \frac{\theta_k}{2}) = \sqrt{\frac{1}{2}(1 - \cos \theta_{k+1})} \Big/ \sqrt{\frac{1}{2}(1 - \cos \theta_k)}$$

$$\leq \left[ \left( 1 - \frac{h^{\frac{1}{2}}(\theta_k)(\eta\lambda + \cos \theta_k) - 2\eta M \sin \frac{\theta_k}{2} \sin \theta_k}{1 + \eta^2\lambda^2 + 2\eta\lambda \cos \theta_k} \right) \Big/ (1 - \cos \theta_k) \right]^{\frac{1}{2}},$$

where

$$h(\theta_k) = 1 + \eta^2\lambda^2 + 2\eta\lambda \cos \theta_k - 4\eta^2 M^2 \sin^2 \frac{\theta_k}{2}.$$

Denote

$$f_1(\theta_k) := \left( 1 - \frac{h^{\frac{1}{2}}(\theta_k)(\eta\lambda + \cos \theta_k) - 2\eta M \sin \frac{\theta_k}{2} \sin \theta_k}{1 + \eta^2\lambda^2 + 2\eta\lambda \cos \theta_k} \right) \Big/ (1 - \cos \theta_k). \tag{37}$$

Because $f_1(\theta_k)$ and its result after L'Hospital's rule are both the type of "$\frac{0}{0}$" when $\theta_k$ tends to 0, we apply L'Hospital's rule twice for $f_1(\theta_k)$ and obtain the following lemma:

**Lemma C.2.** *Suppose $f_1(\theta_k)$ is defined as (37) with $\eta, \lambda, M > 0$. Then, we have:*

$$\lim_{\theta_k \to 0} f_1(\theta_k) = \left( \frac{1 + \eta M}{1 + \eta\lambda} \right)^2. \tag{38}$$

See proof in Appendix C.5.1.

Since $\lambda > UM$ with $U \geq 2$, if $\theta_k$ tends to 0,

$$\frac{d_{k+1}}{d_k} \leq \sqrt{\left( \frac{1 + \eta M}{1 + \eta\lambda} \right)^2} < \frac{1 + \eta M}{1 + U\eta M} < 1.$$

$\forall \sigma \in (0, \frac{\eta M}{2 + 2U\eta M}), \exists \theta \in (0, \frac{\pi}{3})$, if $\theta_k \in (0, \theta)$, namely, $\boldsymbol{\delta}_k \in \mathbb{S}(\mathbf{0}, \epsilon) \cap \mathbb{B}(\boldsymbol{\delta}^\triangle, 2\epsilon \sin \frac{\theta}{2})$,

$$\frac{d_{k+1}}{d_k} < \frac{1 + \eta M}{1 + U\eta M} + \sigma < 1. \tag{39}$$

Next, we prove that (39) holds for $\forall k \in \mathbb{Z}^+$ by mathematical induction. Suppose $\epsilon < \frac{\|\nabla\Phi(\mathbf{0})\|}{(U+1)M}$ and $\boldsymbol{\delta}_0 \in \mathbb{S}(\mathbf{0}, \epsilon) \cap \mathbb{B}(\boldsymbol{\delta}^\triangle, 2\epsilon \sin \frac{\theta}{2})$, then

$$\frac{d_1}{d_0} < \frac{1 + \eta M}{1 + U\eta M} + \sigma < 1. \tag{40}$$

For $k = 1$, according to Lemma C.1 and formulations (36), (39) and (40),

$$\theta_k = 2\arcsin \frac{d_k}{2}, d_1 < d_0 \Rightarrow 0 < \theta_1 < \theta_0 < \theta,$$

and

$$\frac{d_2}{d_1} < \frac{1 + \eta M}{1 + U\eta M} + \sigma < 1.$$

Thus, (39) holds for $k = 1$. Assume (39) holds for $k = j$ where $j \in \mathbb{Z}^+$, namely,

$$\frac{d_{j+1}}{d_j} < \frac{1 + \eta M}{1 + U\eta M} + \sigma < 1. \tag{41}$$

Based on Lemma C.1 and formulations (36), (39) and (41),

$$\theta_k = 2\arcsin \frac{d_k}{2}, d_{j+1} < d_j \Rightarrow 0 < \theta_{j+1} < \theta_j < \cdots < \theta_0 < \theta,$$

and

$$\frac{d_{j+2}}{d_{j+1}} < \frac{1+\eta M}{1+U\eta M} + \sigma < 1.$$

Thus, (39) still holds for $k = j + 1$.

To sum up, if $\epsilon < \frac{\|\nabla\Phi(\mathbf{0})\|}{(U+1)M}$, there exists $\theta \in (0, \frac{\pi}{3})$, when $\boldsymbol{\delta}_0 \in \mathbb{S}(\mathbf{0}, \epsilon) \cap \mathbb{B}(\boldsymbol{\delta}^\triangle, 2\epsilon\sin\frac{\theta}{2})$, we have

$$\|\boldsymbol{\delta}_k - \boldsymbol{\delta}^\triangle\| \le \left(\frac{1+\eta M}{1+U\eta M} + \sigma\right)^k \|\boldsymbol{\delta}_0 - \boldsymbol{\delta}^\triangle\| \le C_2^k \|\boldsymbol{\delta}_0 - \boldsymbol{\delta}^\triangle\|,$$

where $C_2 = \frac{2+3\eta M}{2+2U\eta M} < 1$ and $U \ge 2$. $\qquad\square$

### C.3    PROOF OF LOWER BOUND

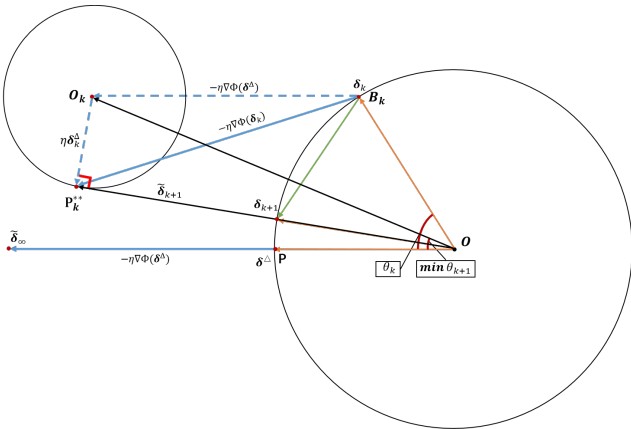

Figure 10: The situation that $\theta_{k+1}$ gets minimum in Figure 8. In this situation, $OP_k$ is externally tangential to $\mathbb{B}(O_k, 2\eta M\epsilon\sin\frac{\theta_k}{2})$ at $P_k^{**}$. Meanwhile, lines $OP$, $OO_k$, and $OP_k^{**}$ are on the same plane.

In this section, we present the proof of the lower bound of the convergence rate for PGD. Following the analysis of (28), the situation that $\theta_{k+1}$ gets minimum is shown in Figure 10 where $P_k = P_k^{**}$. The proof of the lower bound is similar to the proof of the upper bound. Based on formulation (23) and Figure 10,

$$\theta_{k+1} = \angle P_k OP \ge |\angle O_k OP - \angle P_k^{**} OO_k|, \qquad (42)$$

Based on (42) and combine (30) $\sim$ (34),

$$\begin{aligned}
\cos\theta_{k+1} &\le \cos\left(\angle O_k OP - \angle P_k^{**} OO_k\right) \\
&= \cos\angle P_k^{**} OO_k \cos\angle O_k OP + \sin\angle P_k^{**} OO_k \sin\angle O_k OP \\
&= \Big[(1 + \eta^2\lambda^2 + 2\eta\lambda\cos\theta_k - 4\eta^2 M^2\sin^2\frac{\theta_k}{2})^{\frac{1}{2}}(\eta\lambda + \cos\theta_k) \\
&\quad + 2\eta M\sin\frac{\theta_k}{2}\sin\theta_k\Big] \Big/ (1 + \eta^2\lambda^2 + 2\eta\lambda\cos\theta_k).
\end{aligned} \qquad (43)$$

According to (36) and (43), we have

$$\begin{aligned}
\frac{d_{k+1}}{d_k} &= 2\epsilon\sin\frac{\theta_{k+1}}{2} \Big/ (2\epsilon\sin\frac{\theta_k}{2}) = \sqrt{\frac{1}{2}(1 - \cos\theta_{k+1})} \Big/ \sqrt{\frac{1}{2}(1 - \cos\theta_k)} \\
&\ge \left[\left(1 - \frac{h^{\frac{1}{2}}(\theta_k)(\eta\lambda + \cos\theta_k) + 2\eta M\sin\frac{\theta_k}{2}\sin\theta_k}{1 + \eta^2\lambda^2 + 2\eta\lambda\cos\theta_k}\right) \Big/ (1 - \cos\theta_k)\right]^{\frac{1}{2}},
\end{aligned}$$

where $h(\theta_k) = 1 + \eta^2\lambda^2 + 2\eta\lambda\cos\theta_k - 4\eta^2 M^2 \sin^2\frac{\theta_k}{2}$. Denote

$$f_2(\theta_k) := \left(1 - \frac{h^{\frac{1}{2}}(\theta_k)(\eta\lambda + \cos\theta_k) + 2\eta M \sin\frac{\theta_k}{2}\sin\theta_k}{1 + \eta^2\lambda^2 + 2\eta\lambda\cos\theta_k}\right)\bigg/(1 - \cos\theta_k). \tag{44}$$

Because $f_2(\theta_k)$ and its result after L'Hospital's rule are also both the type of "$\frac{0}{0}$" when $\theta_k$ tends to 0, we apply L'Hospital's rule twice for $f_2(\theta_k)$ and obtain the following lemma:

**Lemma C.3.** *Suppose $f_2(\theta_k)$ is defined as (44) with $\eta, \lambda, M > 0$. Then, we have:*

$$\lim_{\theta_k\to 0} f_2(\theta_k) = \left(\frac{1-\eta M}{1+\eta\lambda}\right)^2. \tag{45}$$

See proof in Appendix C.5.2.

Since $\lambda > UM$ and $\eta < \frac{1}{2M}$ where $U \geq 2$, if $\theta_k$ tends to 0,

$$\frac{d_{k+1}}{d_k} \geq \sqrt{\left(\frac{1-\eta M}{1+\eta\lambda}\right)^2} > \frac{1-\eta M}{1+U\eta M} > 0.$$

Then, based on Theorem 3.1, $\forall\sigma \in (0, \frac{\eta M}{2+2U\eta M})$, $\exists\theta \in (0, \frac{\pi}{3})$, if $\theta_k \in (0, \theta)$, namely, $\boldsymbol{\delta}_k \in \mathbb{S}(\mathbf{0}, \epsilon) \cap \mathbb{B}(\boldsymbol{\delta}^\triangle, 2\epsilon\sin\frac{\theta}{2})$,

$$1 > \frac{d_{k+1}}{d_k} > \frac{1-\eta M}{1+U\eta M} - \sigma > 0. \tag{46}$$

Next, we prove that (46) holds for $\forall k \in \mathbb{Z}^+$ by mathematical induction, which is similar to the proof of (38). Suppose $\epsilon < \frac{\|\nabla\Phi(\mathbf{0})\|}{(U+1)M}$ and $\boldsymbol{\delta}_0 \in \mathbb{S}(\mathbf{0}, \epsilon) \cap \mathbb{B}(\boldsymbol{\delta}^\triangle, 2\epsilon\sin\frac{\theta}{2})$, then

$$1 > \frac{d_1}{d_0} > \frac{1-\eta M}{1+U\eta M} - \sigma > 0. \tag{47}$$

For $k = 1$, according to Lemma C.1 and formulations (36), (40) and (47),

$$d_1 < d_0 \Rightarrow 0 < \theta_1 < \theta_0 < \theta,$$

and

$$1 > \frac{d_2}{d_1} > \frac{1-\eta M}{1+U\eta M} - \sigma > 0.$$

Thus, (46) holds for $k = 1$. Assume (46) holds for $k = j$ where $j \in \mathbb{Z}^+$, namely,

$$1 > \frac{d_{j+1}}{d_j} > \frac{1-\eta M}{1+U\eta M} - \sigma > 0. \tag{48}$$

Based on Lemma C.1 and formulations (36), (41) and (48),

$$d_{j+1} < d_j \Rightarrow 0 < \theta_{j+1} < \theta_j < \cdots < \theta_0 < \theta,$$

and

$$1 > \frac{d_{j+2}}{d_{j+1}} > \frac{1-\eta M}{1+U\eta M} - \sigma > 0.$$

Thus, (46) still holds for $k = j + 1$.

To sum up, if $\epsilon < \frac{\|\nabla\Phi(\mathbf{0})\|}{(U+1)M}$ and $\eta < \frac{1}{2M}$, there exists $\theta \in (0, \frac{\pi}{3})$, when $\boldsymbol{\delta}_0 \in \mathbb{S}(\mathbf{0}, \epsilon) \cap \mathbb{B}(\boldsymbol{\delta}^\triangle, 2\epsilon\sin\frac{\theta}{2})$, we have

$$\|\boldsymbol{\delta}_k - \boldsymbol{\delta}^\triangle\| \geq \left(\frac{1-\eta M}{1+U\eta M} - \sigma\right)^k \|\boldsymbol{\delta}_0 - \boldsymbol{\delta}^\triangle\| \geq C_3^k \|\boldsymbol{\delta}_0 - \boldsymbol{\delta}^\triangle\|,$$

where $C_3 = \frac{2-3\eta M}{2+2U\eta M} < C_2 < 1$ and $U \geq 2$.

$\square$

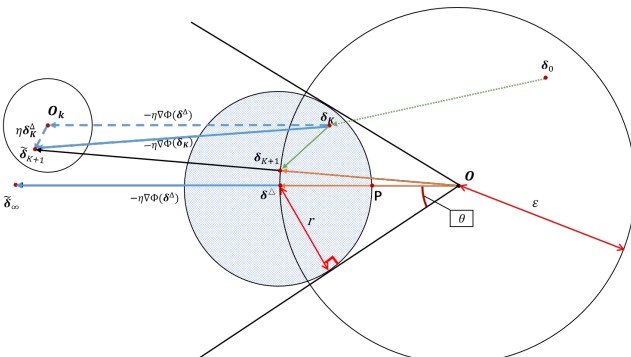

Figure 11: The process of PGD: $\boldsymbol{\delta}_K \to \tilde{\boldsymbol{\delta}}_{K+1} \to \boldsymbol{\delta}_{K+1}$. The Light blue filled ball represents $\mathbb{B}(\boldsymbol{\delta}^\triangle, r)$. The green trajectory of $\boldsymbol{\delta}_k$ is generated by PGD. The solid blue arrow means $-\eta \nabla \Phi(\boldsymbol{\delta}_K)$. The dotted blue arrows indicate auxiliary vectors $-\eta \nabla \Phi(\boldsymbol{\delta}^\triangle)$ and $\eta \boldsymbol{\delta}_k^\triangle$, where $\eta < \frac{1}{2M}$ and $\boldsymbol{\delta}_k^\triangle := \nabla \Phi(\boldsymbol{\delta}^\triangle) - \nabla \Phi(\boldsymbol{\delta}_k)$ with $\|\boldsymbol{\delta}_k^\triangle\| \leq Mr$ (smooth).

### C.4 PROOF OF ARBITRARY INITIALIZATION IN GENERAL NEIGHBORHOOD

Theorem 3.1 shows that there exists $\theta > 0$ such that if the initialization is chosen from $\mathbb{S}(\boldsymbol{0}, \epsilon) \cap \mathbb{B}(\boldsymbol{\delta}^\triangle, 2\epsilon \sin \frac{\theta}{2})$, then PGD exhibits local linear convergence. We now show that PGD achieves global convergence and eventually at least local linear convergence for general initialization.

When the initial point $\boldsymbol{\delta}_0$ is randomly chosen, prior works (Nesterov, 2018; Boyd & Vandenberghe, 2014; Bubeck, 2015) show that the sequence $\boldsymbol{\delta}_k$ generated by PGD converges to the optimal point $\boldsymbol{\delta}^\triangle$ at a global sublinear rate. Therefore, for any $r > 0$, there exists $K(r) \in \mathbb{Z}^+$ such that $\boldsymbol{\delta}_{K+i} \in \mathbb{B}(\boldsymbol{\delta}^\triangle, r) \cap \mathbb{B}(\boldsymbol{0}, \epsilon)$ for all $i \in \mathbb{Z}^+$, where this process is shown in Figure 11.

Next, we specify a suitable choice of $r$ to ensure that $\boldsymbol{\delta}_{K+1}$ satisfies the initialization condition of Theorem 3.1, which guarantees that the sequence $\boldsymbol{\delta}_{K+i}$ for $i = 1, 2, \ldots$ converges at a linear rate.

**1)** To ensure $\tilde{\boldsymbol{\delta}}_{K+1} \notin \mathbb{B}(\boldsymbol{0}, \epsilon)$, in the worst case (when $\boldsymbol{\delta}_K$ is moved to point $P$), we require:

$$\epsilon - r + \eta \|\nabla \Phi(\boldsymbol{\delta}^\triangle)\| - \eta \|\boldsymbol{\delta}_K^\triangle\| > \epsilon,$$

which leads to the sufficient condition:

$$\epsilon - r + \eta \|\nabla \Phi(\boldsymbol{\delta}^\triangle)\| - Mr\eta > \epsilon \Rightarrow r < \frac{\eta \|\nabla \Phi(\boldsymbol{\delta}^\triangle)\|}{M\eta + 1}. \tag{49}$$

**2)** To ensure $\tilde{\boldsymbol{\delta}}_{K+1} \in \left\{ \tilde{\boldsymbol{\delta}} \mid 0 \leq \angle(\tilde{\boldsymbol{\delta}}, \boldsymbol{\delta}^\triangle) < \theta \right\}$, since $\eta \|\boldsymbol{\delta}_k^\triangle\| \leq \eta Mr < r$, it is sufficient that:

$$r < \epsilon \sin \theta.$$

**3)** We also require $\tilde{\boldsymbol{\delta}}_{K+1} \notin \mathbb{B}(\boldsymbol{\delta}^\triangle, r)$, whose sufficient condition is:

$$\eta \|\nabla \Phi(\boldsymbol{\delta}_K)\| > \eta(\|\nabla \Phi(\boldsymbol{\delta}^\triangle)\| - Mr) > 2r \Rightarrow r < \frac{\eta \|\nabla \Phi(\boldsymbol{\delta}^\triangle)\|}{M\eta + 2} \tag{50}$$

**Conclusion:** For any

$$r < \min \left\{ \epsilon \sin \theta, \frac{\eta \|\nabla \Phi(\boldsymbol{\delta}^\triangle)\|}{M\eta + 2} \right\},$$

there exists $K \in \mathbb{Z}^+$ such that

$$\tilde{\boldsymbol{\delta}}_{K+1} \in \left\{ \tilde{\boldsymbol{\delta}} \mid 0 \leq \angle(\tilde{\boldsymbol{\delta}}, \boldsymbol{\delta}^\triangle) < \theta, \; \tilde{\boldsymbol{\delta}} \notin \mathbb{B}(\boldsymbol{0}, \epsilon) \right\}.$$

If $\boldsymbol{\delta}_{K+1} \neq \boldsymbol{\delta}^\triangle$, then it satisfies the initialization condition of Theorem 3.1, and the subsequent sequence $\boldsymbol{\delta}_{K+i}$ for $i = 1, 2, \ldots$ converges at least at a linear rate.

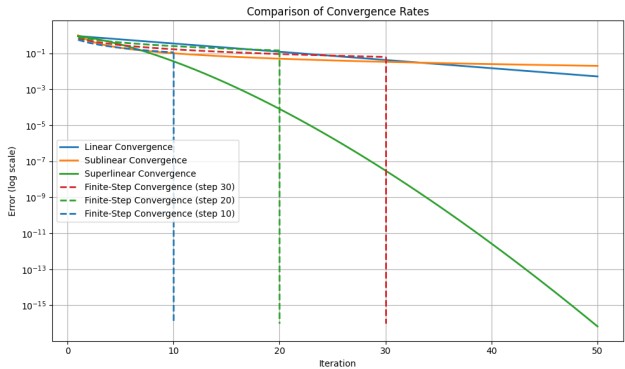

Figure 12: Comparison of convergence rates for optimization algorithms. The figure illustrates typical error decay behaviors: linear convergence (exponential decay), sublinear convergence (e.g., $1/k$), and superlinear convergence (faster than exponential). Additionally, three finite-step convergence scenarios are shown, where error drops to zero suddenly after a fixed number of iterations (at step 10, 20, and 30 respectively), representing idealized convergence in a finite number of steps. The $y$-axis is shown on a logarithmic scale to highlight differences in decay speed.

If $\boldsymbol{\delta}_{K+1} = \boldsymbol{\delta}^{\triangle}$, the sequence $\boldsymbol{\delta}_k$ exhibits finite-step convergence. Prior works (Nocedal & J. Wright, 2006; Trefethen & Bau, 2022; Han et al., 2022) show that such convergence is effectively instantaneous and can be considered faster than linear (See Figure 12). This does not contradict our conclusion, as our paper emphasizes that PGD converges at least at a linear rate, and the lower bound applies under the specific initialization assumed in Theorem 3.1.

In summary, PGD achieves global convergence and eventually at least local linear convergence even under general initialization.

### C.5    SPECIFIC PROOF OF LEMMAS C.2 AND C.3

#### C.5.1    PROOF OF $\lim_{\theta_k \to 0} f_1(\theta_k)$ FOR LEMMA C.2

Suppose $h(\theta_k) = 1 + \eta^2\lambda^2 + 2\eta\lambda\cos\theta_k - 4\eta^2 M^2 \sin^2\frac{\theta_k}{2}$. Then,

$$h(\theta_k) = 1 + \eta^2\lambda^2 + 2\eta\lambda\cos\theta_k - 4\eta^2 M^2 \sin^2\frac{\theta_k}{2},$$

$$h'(\theta_k) = -2(\eta\lambda + \eta^2 M^2)\sin\theta_k,$$

$$h''(\theta_k) = -2(\eta\lambda + \eta^2 M^2)\cos\theta_k.$$

And

$$h(0) = (1 + \eta\lambda)^2, h'(0) = 0, h''(0) = -2(\eta\lambda + \eta^2 M^2). \tag{51}$$

Denote

$$f_1(\theta_k) := \big(1 - \frac{h^{\frac{1}{2}}(\theta_k)(\eta\lambda + \cos\theta_k) - 2\eta M \sin\frac{\theta_k}{2}\sin\theta_k}{1 + \eta^2\lambda^2 + 2\eta\lambda\cos\theta_k}\big)\Big/(1 - \cos\theta_k). \tag{52}$$

Based on (51),

$$\lim_{\theta_k \to 0} f_1(\theta_k) = \lim_{\theta_k \to 0} \big(1 - \frac{h(\theta_k)^{\frac{1}{2}}(\eta\lambda + \cos\theta_k) - 2\eta M \sin\frac{\theta_k}{2}\sin\theta_k}{(1 + \eta\lambda)^2 + 2\eta\lambda\cos\theta_k}\big)\Big/(1 - \cos\theta_k)$$

$$= \lim_{\theta_k \to 0} \big(1 - \frac{(1 + \eta\lambda)(\eta\lambda + 1)}{1 + \eta^2\lambda^2 + 2\eta\lambda}\big)\Big/(1 - 1)$$

$$= \frac{0}{0}.$$

Because $f_1(\theta_k)$ is the type of "$\frac{0}{0}$" when $\theta_k$ tends to 0, we apply L'Hospital's rule for (52).

$$
\lim_{\theta_k \to 0} f_1(\theta_k) = \lim_{\theta_k \to 0} \Big(1 - \frac{h(\theta_k)^{\frac{1}{2}}(\eta\lambda + \cos\theta_k) - 2\eta M \sin\frac{\theta_k}{2}\sin\theta_k}{1 + \eta^2\lambda^2 + 2\eta\lambda\cos\theta_k}\Big)' \Big/ (1 - \cos\theta_k)'
$$

$$
= \lim_{\theta_k \to 0} \Big\{ \Big[\frac{1}{2}h^{-\frac{1}{2}}(\theta_k)h'(\theta_k)(\eta\lambda + \cos\theta_k) - h(\theta_k)^{\frac{1}{2}}\sin\theta_k - 2\eta M(\frac{1}{2}\cos\frac{\theta_k}{2}\sin\theta_k
$$

$$
+ \sin\frac{\theta_k}{2}\cos\theta_k)\Big](1 + \eta^2\lambda^2 + 2\eta\lambda\cos\theta_k) + 2\eta\lambda\sin\theta_k\Big[h^{\frac{1}{2}}(\theta_k)(\eta\lambda + \cos\theta_k) -
$$

$$
2\eta M \sin\frac{\theta_k}{2}\sin\theta_k\Big]\Big\} \Big/ \Big[-\sin\theta_k(1 + \eta^2\lambda^2 + 2\eta\lambda\cos\theta_k)^2\Big]
$$

$$
= \lim_{\theta_k \to 0} \Big\{ \Big[\frac{1}{2}h^{-\frac{1}{2}}(0)h'(0)(\eta\lambda + 1) - h(0)^{\frac{1}{2}}\cdot 0 - 2\eta M \cdot 0\Big](1 + \eta\lambda)^2 +
$$

$$
2\eta\lambda \cdot 0 \cdot \Big[h^{\frac{1}{2}}(0)(\eta\lambda + 1) - 0\Big]\Big\} \Big/ \Big[0 \cdot (1 + \eta\lambda)^4\Big]
$$

$$
= \frac{0}{0}.
$$

$f_1(\theta_k)$ is still the type of "$\frac{0}{0}$" after using L'Hospital's rule when $\theta_k$ tends to 0. Then, we apply L'Hospital's rule again for $f_1(\theta_k)$ for (52),

$$
\lim_{\theta_k \to 0} f_1(\theta_k) = \lim_{\theta_k \to 0} \Big\{ \Big[\frac{1}{2}h^{-\frac{1}{2}}(\theta_k)h'(\theta_k)(\eta\lambda + \cos\theta_k) - h(\theta_k)^{\frac{1}{2}}\sin\theta_k - 2\eta M(\frac{1}{2}\cos\frac{\theta_k}{2}\sin\theta_k
$$

$$
+ \sin\frac{\theta_k}{2}\cos\theta_k)\Big](1 + \eta^2\lambda^2 + 2\eta\lambda\cos\theta_k) + 2\eta\lambda\sin\theta_k\Big[h^{\frac{1}{2}}(\theta_k)(\eta\lambda + \cos\theta_k) -
$$

$$
2\eta M \sin\frac{\theta_k}{2}\sin\theta_k\Big]\Big\}' \Big/ \Big[-\sin\theta_k(1 + \eta^2\lambda^2 + 2\eta\lambda\cos\theta_k)^2\Big]'
$$

$$
= \lim_{\theta_k \to 0} \Big\{ \Big[-\frac{1}{4}h^{-\frac{3}{2}}(\theta_k)(h'(\theta_k))^2(\eta\lambda + \cos\theta_k) + \frac{1}{2}h^{-\frac{1}{2}}(\theta_k)h''(\theta_k)(\eta\lambda + \cos\theta_k) -
$$

$$
\frac{1}{2}h^{-\frac{1}{2}}(\theta_k)h'(\theta_k)\sin\theta_k - \frac{1}{2}h^{-\frac{1}{2}}(\theta_k)h'(\theta_k)\sin\theta_k - h^{\frac{1}{2}}(\theta_k)\cos\theta_k -
$$

$$
2\eta M(-\frac{1}{4}\sin\frac{\theta_k}{2}\sin\theta_k + \frac{1}{2}\cos\frac{\theta_k}{2}\cos\theta_k + \frac{1}{2}\cos\frac{\theta_k}{2}\cos\theta_k - \sin\frac{\theta_k}{2}\sin\theta_k)\Big]
$$

$$
(1 + \eta^2\lambda^2 + 2\eta\lambda\cos\theta_k) - 2\eta\lambda\sin\theta_k\Big[\frac{1}{2}h^{-\frac{1}{2}}(\theta_k)h'(\theta_k)(\eta\lambda + \cos\theta_k) -
$$

$$
h(\theta_k)^{\frac{1}{2}}\sin\theta_k - 2\eta M(\frac{1}{2}\cos\frac{\theta_k}{2}\sin\theta_k + \sin\frac{\theta_k}{2}\cos\theta_k)\Big] + 2\eta\lambda\sin\theta_k \cdot
$$

$$
\Big[\frac{1}{2}h^{-\frac{1}{2}}(\theta_k)h'(\theta_k)(\eta\lambda + \cos\theta_k) - h(\theta_k)^{\frac{1}{2}}\sin\theta_k - 2\eta M(\frac{1}{2}\cos\frac{\theta_k}{2}\sin\theta_k +
$$

$$
\sin\frac{\theta_k}{2}\cos\theta_k)\Big] + 2\eta\lambda\cos\theta_k\Big[h^{\frac{1}{2}}(\theta_k)(\eta\lambda + \cos\theta_k) - 2\eta M \sin\frac{\theta_k}{2}\sin\theta_k\Big]\Big\}
$$

$$
\Big/ \Big[4\eta\lambda\sin^2\theta_k(1 + \eta^2\lambda^2 + 2\eta\lambda\cos\theta_k) - \cos\theta_k(1 + \eta^2\lambda^2 + 2\eta\lambda\cos\theta_k)^2\Big]
$$

$$
= \lim_{\theta_k \to 0} \Bigg\{ \Big[ -\frac{1}{4} h^{-\frac{1}{2}}(\theta_k)(h'(\theta_k))^2(\eta\lambda + \cos\theta_k) + \frac{1}{2} h^{-\frac{1}{2}}(\theta_k) h''(\theta_k)(\eta\lambda + \cos\theta_k) -
$$

$$
h^{-\frac{1}{2}}(\theta_k) h'(\theta_k) \sin\theta_k - h^{\frac{1}{2}}(\theta_k)\cos\theta_k - 2\eta M \Big(-\frac{5}{4}\sin\frac{\theta_k}{2}\sin\theta_k + \cos\frac{\theta_k}{2}\cos\theta_k\Big) \Big]
$$

$$
(1 + \eta^2\lambda^2 + 2\eta\lambda\cos\theta_k) + 2\eta\lambda\cos\theta_k \Big[ h^{\frac{1}{2}}(\theta_k)(\eta\lambda + \cos\theta_k) - 2\eta M \sin\frac{\theta_k}{2}\sin\theta_k \Big] \Bigg\}
$$

$$
\Big/ \Big[ 4\eta\lambda\sin^2\theta_k(1 + \eta^2\lambda^2 + 2\eta\lambda\cos\theta_k) - \cos\theta_k(1 + \eta^2\lambda^2 + 2\eta\lambda\cos\theta_k)^2 \Big]
$$

$$
= \Bigg\{ \Big[ -\frac{1}{4} h^{-\frac{1}{2}}(0)(h'(0))^2(\eta\lambda + 1) + \frac{1}{2} h^{-\frac{1}{2}}(0) h''(0)(\eta\lambda + 1) - h^{-\frac{1}{2}}(0) h'(0) \cdot 0
$$

$$
- h^{\frac{1}{2}}(0) - 2\eta M(0 + 1) \Big](1 + \eta^2\lambda^2 + 2\eta\lambda) + 2\eta\lambda \Big[ h^{\frac{1}{2}}(0)(\eta\lambda + 1) - 2\eta M \cdot 0 \Big] \Bigg\}
$$

$$
\Big/ \Big[ 0 \cdot (1 + \eta^2\lambda^2 + 2\eta\lambda) - (1 + \eta^2\lambda^2 + 2\eta\lambda)^2 \Big]
$$

$$
= \frac{\Big[ \frac{1}{2}(1+\eta\lambda)^{-1}\cdot[-2(\eta\lambda+\eta^2 M^2)](\eta\lambda+1) - (1+\eta\lambda) - 2\eta M \Big](1+\eta\lambda)^2 + 2\eta\lambda(1+\eta\lambda)^2}{-(1+\eta\lambda)^4}
$$

$$
= \Big(\frac{1 + \eta M}{1 + \eta\lambda}\Big)^2.
$$

□

### C.5.2 Proof of $\lim_{\theta_k \to 0} f_2(\theta_k)$ for Lemma C.3

Suppose $h(\theta_k) = 1 + \eta^2\lambda^2 + 2\eta\lambda\cos\theta_k - 4\eta^2 M^2 \sin^2\frac{\theta_k}{2}$ Denote

$$
f_2(\theta_k) := \Big(1 - \frac{h^{\frac{1}{2}}(\theta_k)(\eta\lambda + \cos\theta_k) + 2\eta M \sin\frac{\theta_k}{2}\sin\theta_k}{1 + \eta^2\lambda^2 + 2\eta\lambda\cos\theta_k}\Big) \Big/ (1 - \cos\theta_k). \tag{53}
$$

Based on (51),

$$
\lim_{\theta_k \to 0} f_2(\theta_k) = \lim_{\theta_k \to 0} \Big(1 - \frac{h(\theta_k)^{\frac{1}{2}}(\eta\lambda + \cos\theta_k) + 2\eta M \sin\frac{\theta_k}{2}\sin\theta_k}{(1 + \eta\lambda)^2 + 2\eta\lambda\cos\theta_k}\Big) \Big/ (1 - \cos\theta_k)
$$

$$
= \lim_{\theta_k \to 0} \Big(1 - \frac{(1 + \eta\lambda)(\eta\lambda + 1)}{1 + \eta^2\lambda^2 + 2\eta\lambda}\Big) \Big/ (1 - 1)
$$

$$
= \frac{0}{0}.
$$

Since $f_2(\theta_k)$ is the type of "$\frac{0}{0}$" when $\theta_k$ tends to 0, we apply L'Hospital's rule for (53).

$$\lim_{\theta_k \to 0} f_2(\theta_k) = \lim_{\theta_k \to 0} \left(1 - \frac{h(\theta_k)^{\frac{1}{2}}(\eta\lambda + \cos\theta_k) + 2\eta M \sin\frac{\theta_k}{2}\sin\theta_k}{1 + \eta^2\lambda^2 + 2\eta\lambda\cos\theta_k}\right)' \Big/ (1 - \cos\theta_k)'$$

$$= \lim_{\theta_k \to 0} \left\{\left[\frac{1}{2}h^{-\frac{1}{2}}(\theta_k)h'(\theta_k)(\eta\lambda + \cos\theta_k) - h(\theta_k)^{\frac{1}{2}}\sin\theta_k + 2\eta M(\frac{1}{2}\cos\frac{\theta_k}{2}\sin\theta_k\right.\right.$$

$$\left.+ \sin\frac{\theta_k}{2}\cos\theta_k)\right](1 + \eta^2\lambda^2 + 2\eta\lambda\cos\theta_k) + 2\eta\lambda\sin\theta_k\left[h^{\frac{1}{2}}(\theta_k)(\eta\lambda + \cos\theta_k)+\right.$$

$$\left.\left.2\eta M\sin\frac{\theta_k}{2}\sin\theta_k\right]\right\} \Big/ \left[-\sin\theta_k(1 + \eta^2\lambda^2 + 2\eta\lambda\cos\theta_k)^2\right]$$

$$= \lim_{\theta_k \to 0} \left\{\left[\frac{1}{2}h^{-\frac{1}{2}}(0)h'(0)(\eta\lambda + 1) - h(0)^{\frac{1}{2}} \cdot 0 + 2\eta M \cdot 0\right](1 + \eta\lambda)^2+\right.$$

$$\left.2\eta\lambda \cdot 0 \cdot \left[h^{\frac{1}{2}}(0)(\eta\lambda + 1) + 0\right]\right\} \Big/ \left[0 \cdot (1 + \eta\lambda)^4\right]$$

$$= \frac{0}{0}.$$

Because $f_2(\theta_k)$ and its result after L'Hospital's rule are also both the type of "$\frac{0}{0}$" when $\theta_k$ tends to 0, we apply L'Hospital's rule twice for $f_2(\theta_k)$,

$$\lim_{\theta_k \to 0} f_2(\theta_k) = \lim_{\theta_k \to 0} \left\{\left[\frac{1}{2}h^{-\frac{1}{2}}(\theta_k)h'(\theta_k)(\eta\lambda + \cos\theta_k) - h(\theta_k)^{\frac{1}{2}}\sin\theta_k + 2\eta M(\frac{1}{2}\cos\frac{\theta_k}{2}\sin\theta_k\right.\right.$$

$$\left.+ \sin\frac{\theta_k}{2}\cos\theta_k)\right](1 + \eta^2\lambda^2 + 2\eta\lambda\cos\theta_k) + 2\eta\lambda\sin\theta_k\left[h^{\frac{1}{2}}(\theta_k)(\eta\lambda + \cos\theta_k)+\right.$$

$$\left.\left.2\eta M\sin\frac{\theta_k}{2}\sin\theta_k\right]\right\}' \Big/ \left[-\sin\theta_k(1 + \eta^2\lambda^2 + 2\eta\lambda\cos\theta_k)^2\right]'$$

$$= \lim_{\theta_k \to 0} \left\{\left[-\frac{1}{4}h^{-\frac{3}{2}}(\theta_k)(h'(\theta_k))^2(\eta\lambda + \cos\theta_k) + \frac{1}{2}h^{-\frac{1}{2}}(\theta_k)h''(\theta_k)(\eta\lambda + \cos\theta_k)+\right.\right.$$

$$\frac{1}{2}h^{-\frac{1}{2}}(\theta_k)h'(\theta_k)\sin\theta_k - \frac{1}{2}h^{-\frac{1}{2}}(\theta_k)h'(\theta_k)\sin\theta_k - h^{\frac{1}{2}}(\theta_k)\cos\theta_k+$$

$$2\eta M(-\frac{1}{4}\sin\frac{\theta_k}{2}\sin\theta_k + \frac{1}{2}\cos\frac{\theta_k}{2}\cos\theta_k + \frac{1}{2}\cos\frac{\theta_k}{2}\cos\theta_k - \sin\frac{\theta_k}{2}\sin\theta_k)\Big]$$

$$(1 + \eta^2\lambda^2 + 2\eta\lambda\cos\theta_k) - 2\eta\lambda\sin\theta_k\left[\frac{1}{2}h^{-\frac{1}{2}}(\theta_k)h'(\theta_k)(\eta\lambda + \cos\theta_k)-\right.$$

$$\left.h(\theta_k)^{\frac{1}{2}}\sin\theta_k + 2\eta M(\frac{1}{2}\cos\frac{\theta_k}{2}\sin\theta_k + \sin\frac{\theta_k}{2}\cos\theta_k)\right] + 2\eta\lambda\sin\theta_k \cdot$$

$$\left[\frac{1}{2}h^{-\frac{1}{2}}(\theta_k)h'(\theta_k)(\eta\lambda + \cos\theta_k) - h(\theta_k)^{\frac{1}{2}}\sin\theta_k + 2\eta M(\frac{1}{2}\cos\frac{\theta_k}{2}\sin\theta_k+\right.$$

$$\left.\sin\frac{\theta_k}{2}\cos\theta_k)\right] + 2\eta\lambda\cos\theta_k\left[h^{\frac{1}{2}}(\theta_k)(\eta\lambda + \cos\theta_k) + 2\eta M\sin\frac{\theta_k}{2}\sin\theta_k\right]\Big\}$$

$$\Big/ \left[4\eta\lambda\sin^2\theta_k(1 + \eta^2\lambda^2 + 2\eta\lambda\cos\theta_k) - \cos\theta_k(1 + \eta^2\lambda^2 + 2\eta\lambda\cos\theta_k)^2\right]$$

$$= \lim_{\theta_k \to 0} \left\{ \left[ -\frac{1}{4}h^{-\frac{1}{2}}(\theta_k)(h'(\theta_k))^2(\eta\lambda + \cos\theta_k) + \frac{1}{2}h^{-\frac{1}{2}}(\theta_k)h''(\theta_k)(\eta\lambda + \cos\theta_k) - \right.\right.$$

$$h^{-\frac{1}{2}}(\theta_k)h'(\theta_k)\sin\theta_k - h^{\frac{1}{2}}(\theta_k)\cos\theta_k + 2\eta M\left(-\frac{5}{4}\sin\frac{\theta_k}{2}\sin\theta_k + \cos\frac{\theta_k}{2}\cos\theta_k\right)\right]$$

$$(1 + \eta^2\lambda^2 + 2\eta\lambda\cos\theta_k) + 2\eta\lambda\cos\theta_k\left[h^{\frac{1}{2}}(\theta_k)(\eta\lambda + \cos\theta_k) + 2\eta M \sin\frac{\theta_k}{2}\sin\theta_k\right]\right\}$$

$$\left/ \left[4\eta\lambda\sin^2\theta_k(1 + \eta^2\lambda^2 + 2\eta\lambda\cos\theta_k) - \cos\theta_k(1 + \eta^2\lambda^2 + 2\eta\lambda\cos\theta_k)^2\right]\right.$$

$$= \left\{\left[ -\frac{1}{4}h^{-\frac{1}{2}}(0)(h'(0))^2(\eta\lambda + 1) + \frac{1}{2}h^{-\frac{1}{2}}(0)h''(0)(\eta\lambda + 1) - h^{-\frac{1}{2}}(0)h'(0) \cdot 0\right.\right.$$

$$\left. - h^{\frac{1}{2}}(0) + 2\eta M(0 + 1)\right](1 + \eta^2\lambda^2 + 2\eta\lambda) + 2\eta\lambda\left[h^{\frac{1}{2}}(0)(\eta\lambda + 1) + 2\eta M \cdot 0\right]\right\}$$

$$\left/ \left[0 \cdot (1 + \eta^2\lambda^2 + 2\eta\lambda) - (1 + \eta^2\lambda^2 + 2\eta\lambda)^2\right]\right.$$

$$= \frac{\left\{\frac{1}{2}(1+\eta\lambda)^{-1} \cdot [-2(\eta\lambda + \eta^2 M^2)](\eta\lambda + 1) - (1 + \eta\lambda) + 2\eta M\right\}(1+\eta\lambda)^2 + 2\eta\lambda(1+\eta\lambda)^2}{-(1+\eta\lambda)^4}$$

$$= \left(\frac{1 - \eta M}{1 + \eta\lambda}\right)^2.$$

$\square$

# D PROOFS IN CONTINUOUS-TIME SCENARIO

## D.1 PROOF OF FORMULATION (7)

$$\boldsymbol{\delta}_{k+1} = \mathcal{P}_{\mathbb{B}(\mathbf{0},\epsilon)}\left[\boldsymbol{\delta}(t_k) - \eta\nabla\Phi(\boldsymbol{\delta}(t_k))\right] = \epsilon\frac{\boldsymbol{\delta}(t_k) - \eta\nabla\Phi(\boldsymbol{\delta}(t_k))}{\|\boldsymbol{\delta}(t_k) - \eta\nabla\Phi(\boldsymbol{\delta}(t_k))\|},$$

$$= \boldsymbol{\delta}(t_k) + \eta\epsilon\left[\frac{\boldsymbol{\delta}(t_k) - \eta\nabla\Phi(\boldsymbol{\delta}(t_k))}{\eta\|\boldsymbol{\delta}(t_k) - \eta\nabla\Phi(\boldsymbol{\delta}(t_k))\|}\right] - \boldsymbol{\delta}(t_k),$$

$$= \boldsymbol{\delta}(t_k) + \eta \cdot \frac{\epsilon}{\|\boldsymbol{\delta}(t_k) - \eta\nabla\Phi(\boldsymbol{\delta}(t_k))\|}\left[\boldsymbol{\delta}(t_k)\frac{(\epsilon - \|\boldsymbol{\delta}(t_k) - \eta\nabla\Phi(\boldsymbol{\delta}(t_k))\|)}{\eta\epsilon} - \nabla\Phi(\boldsymbol{\delta}(t_k))\right], \quad (54)$$

$$= \boldsymbol{\delta}(t_k) + \eta \cdot \frac{\epsilon}{\|\boldsymbol{\delta}(t_k) - \eta\nabla\Phi(\boldsymbol{\delta}(t_k))\|}\left[\boldsymbol{\delta}(t_k)\frac{(\|\boldsymbol{\delta}(t_k)\|^2 - \|\boldsymbol{\delta}(t_k) - \eta\nabla\Phi(\boldsymbol{\delta}(t_k))\|^2)}{\eta\epsilon(\|\boldsymbol{\delta}(t_k)\| + \|\boldsymbol{\delta}(t_k) - \eta\nabla\Phi(\boldsymbol{\delta}(t_k))\|)} - \nabla\Phi(\boldsymbol{\delta}(t_k))\right],$$

$$= \boldsymbol{\delta}(t_k) + \eta \cdot \frac{\epsilon}{\|\boldsymbol{\delta}(t_k) - \eta\nabla\Phi(\boldsymbol{\delta}(t_k))\|}\left[\boldsymbol{\delta}(t_k)\frac{\langle 2\boldsymbol{\delta}(t_k) - \eta\nabla\Phi(\boldsymbol{\delta}(t_k)), \nabla\Phi(\boldsymbol{\delta}(t_k))\rangle}{\epsilon(\|\boldsymbol{\delta}(t_k)\| + \|\boldsymbol{\delta}(t_k) - \eta\nabla\Phi(\boldsymbol{\delta}(t_k))\|)} - \nabla\Phi(\boldsymbol{\delta}(t_k))\right].$$

$\square$

## D.2 PROOF OF THEOREM 4.1

*Theorem* 4.1. Suppose that (i) $\Phi(\boldsymbol{\delta}) \in \mathcal{F}_M$ and $\boldsymbol{\delta}^* \notin \mathbb{B}(\mathbf{0}, \epsilon)$, (ii) $\epsilon < \frac{\|\nabla\Phi(\mathbf{0})\|}{(U+1)M}$ with $U \geq 2$, (iii) $\boldsymbol{\delta}(0) = \boldsymbol{\delta}_0 \in \mathbb{S}(\mathbf{0}, \epsilon) \cap \mathbb{B}(\boldsymbol{\delta}^\triangle, \sqrt{2}\epsilon)$, we have $\boldsymbol{\delta}(t) \in \mathbb{S}(\mathbf{0}, \epsilon)$, where $\boldsymbol{\delta}(t)$ is the solution to ODE (3).

*Proof.* Denote

$$r(t) := \|\boldsymbol{\delta}(t)\|^2 = \langle\boldsymbol{\delta}(t), \boldsymbol{\delta}(t)\rangle. \tag{55}$$

Based on ODE (3) and (55),

$$r'(t) = 2\boldsymbol{\delta}^\top(t)\frac{\partial\boldsymbol{\delta}(t)}{\partial t} = 2\boldsymbol{\delta}^\top(t)\left(\frac{1}{\epsilon^2}\boldsymbol{\delta}(t)\langle\boldsymbol{\delta}(t), \nabla\Phi(\boldsymbol{\delta}(t))\rangle - \nabla\Phi(\boldsymbol{\delta}(t))\right)$$

$$= 2\left(\frac{1}{\epsilon^2}(\boldsymbol{\delta}^\top(t)\boldsymbol{\delta}(t))\boldsymbol{\delta}^\top(t)\nabla\Phi(\boldsymbol{\delta}(t)) - \boldsymbol{\delta}^\top(t)\nabla\Phi(\boldsymbol{\delta}(t))\right)$$

$$= 2\left(\boldsymbol{\delta}^\top(t)\nabla\Phi(\boldsymbol{\delta}(t)) - \boldsymbol{\delta}^\top(t)\nabla\Phi(\boldsymbol{\delta}(t))\right)$$

$$= 0,$$

Because $\boldsymbol{\delta}(0) = \boldsymbol{\delta}_0 \in \mathbb{S}(\mathbf{0}, \epsilon) \cap \mathbb{B}(\boldsymbol{\delta}^\triangle, \sqrt{2}\epsilon)$, we have $r(0) = \boldsymbol{\delta}^2(0) = \epsilon^2$ and $\|\boldsymbol{\delta}(t)\| = \epsilon$, which means that $\boldsymbol{\delta}(t) \subset \mathbb{S}(\mathbf{0}, \epsilon)$. $\qquad\square$

### D.3 Proof of Theorem 4.2

*Theorem* 4.2 (Existence and Uniqueness). Suppose that (i) $\Phi(\boldsymbol{\delta}) \in \mathcal{F}_M$ and $\boldsymbol{\delta}^* \notin \mathbb{B}(\mathbf{0}, \epsilon)$, (ii) $\epsilon < \frac{\|\nabla\Phi(\mathbf{0})\|}{(U+1)M}$ with $U \geq 2$, then the ODE (3) with initial condition $\boldsymbol{\delta}_0 \in \mathbb{S}(\mathbf{0}, \epsilon) \cap \mathbb{B}(\boldsymbol{\delta}^\triangle, \sqrt{2}\epsilon)$ has a unique solution $\boldsymbol{\delta}(t) \in C^1([0, \infty); \mathbb{R}^d)$.

*Proof.* To ensure the existence and uniqueness of the solution to ODE (3), we first introduce the following lemma:

**Lemma D.1.** *(Khalil, 2002, Theorem 3.3) Let $\boldsymbol{F}(\boldsymbol{\delta})$ in ODE (4) be locally Lipshitz for all $\boldsymbol{\delta}$ in a domain $D \subset \mathbb{R}^d$. Let $W$ be a compact subset of $D$, $\boldsymbol{\delta}_0 \in D$, and suppose it is known that every solution of ODE (4) lies entirely in $W$. Then, the ODE (4) has a unique solution that is defined for $t \geq 0$.*

According to Theorem 4.1, the conditions of Theorem 4.2 guarantee $\boldsymbol{\delta}(t) \subset \mathbb{S}(\mathbf{0}, \epsilon)$, which means that the solution of ODE (3) lies entirely in $\mathbb{B}(\mathbf{0}, \epsilon)$. Compared ODE (3) with ODE (4), we define

$$\boldsymbol{F}(\boldsymbol{\delta}) := \frac{1}{\epsilon^2}\boldsymbol{\delta}\langle\boldsymbol{\delta}, \nabla\Phi(\boldsymbol{\delta})\rangle - \nabla\Phi(\boldsymbol{\delta}).$$

Because $\Phi(\boldsymbol{\delta}) \in \mathcal{F}_M$,

$$\|\nabla\Phi(\boldsymbol{\delta}) - \nabla\Phi(\mathbf{0})\| \leq M\|\boldsymbol{\delta} - \mathbf{0}\| \Rightarrow \|\nabla\Phi(\boldsymbol{\delta})\| \leq \|\nabla\Phi(\mathbf{0})\| + M\epsilon.$$

Then,

$$\|\boldsymbol{F}(\widetilde{\boldsymbol{\delta}}) - \boldsymbol{F}(\boldsymbol{\delta})\| = \|\left(\frac{1}{\epsilon^2}\widetilde{\boldsymbol{\delta}}\langle\widetilde{\boldsymbol{\delta}}, \nabla\Phi(\widetilde{\boldsymbol{\delta}})\rangle - \nabla\Phi(\widetilde{\boldsymbol{\delta}})\right) - \left(\frac{1}{\epsilon^2}\boldsymbol{\delta}\langle\boldsymbol{\delta}, \nabla\Phi(\boldsymbol{\delta})\rangle - \nabla\Phi(\boldsymbol{\delta})\right)\|$$

$$\leq \frac{1}{\epsilon^2}\|\boldsymbol{\delta}\boldsymbol{\delta}^\top\nabla\Phi(\boldsymbol{\delta}) - \widetilde{\boldsymbol{\delta}}\widetilde{\boldsymbol{\delta}}^\top\nabla\Phi(\widetilde{\boldsymbol{\delta}})\| + \|\nabla\Phi(\widetilde{\boldsymbol{\delta}}) - \nabla\Phi(\boldsymbol{\delta})\|$$

$$\leq \frac{1}{\epsilon^2}\|\boldsymbol{\delta}\boldsymbol{\delta}^\top\nabla\Phi(\boldsymbol{\delta}) - \boldsymbol{\delta}\boldsymbol{\delta}^\top\nabla\Phi(\widetilde{\boldsymbol{\delta}}) + \boldsymbol{\delta}\boldsymbol{\delta}^\top\nabla\Phi(\widetilde{\boldsymbol{\delta}}) - \widetilde{\boldsymbol{\delta}}\widetilde{\boldsymbol{\delta}}^\top\nabla\Phi(\widetilde{\boldsymbol{\delta}})\| + M\|\widetilde{\boldsymbol{\delta}} - \boldsymbol{\delta}\|$$

$$\leq \frac{1}{\epsilon^2}\|\boldsymbol{\delta}\boldsymbol{\delta}^\top\nabla\Phi(\boldsymbol{\delta}) - \boldsymbol{\delta}\boldsymbol{\delta}^\top\nabla\Phi(\widetilde{\boldsymbol{\delta}})\| + \frac{1}{\epsilon^2}\|\boldsymbol{\delta}\boldsymbol{\delta}^\top\nabla\Phi(\widetilde{\boldsymbol{\delta}}) - \widetilde{\boldsymbol{\delta}}\widetilde{\boldsymbol{\delta}}^\top\nabla\Phi(\widetilde{\boldsymbol{\delta}})\| + M\|\widetilde{\boldsymbol{\delta}} - \boldsymbol{\delta}\|$$

$$\leq \frac{1}{\epsilon^2}\|\boldsymbol{\delta}\boldsymbol{\delta}^\top\| \cdot \|\nabla\Phi(\boldsymbol{\delta}) - \nabla\Phi(\widetilde{\boldsymbol{\delta}})\| + \frac{1}{\epsilon^2}\|\boldsymbol{\delta}\boldsymbol{\delta}^\top - \widetilde{\boldsymbol{\delta}}\widetilde{\boldsymbol{\delta}}^\top\| \cdot \|\nabla\Phi(\widetilde{\boldsymbol{\delta}})\| + M\|\widetilde{\boldsymbol{\delta}} - \boldsymbol{\delta}\|$$

$$\leq M\|\widetilde{\boldsymbol{\delta}} - \boldsymbol{\delta}\| + \frac{1}{\epsilon^2}(\|\nabla\Phi(\mathbf{0})\| + M\epsilon) \cdot \|\boldsymbol{\delta}\boldsymbol{\delta}^\top - \boldsymbol{\delta}\widetilde{\boldsymbol{\delta}}^\top + \boldsymbol{\delta}\widetilde{\boldsymbol{\delta}}^\top - \widetilde{\boldsymbol{\delta}}\widetilde{\boldsymbol{\delta}}^\top\| + M\|\widetilde{\boldsymbol{\delta}} - \boldsymbol{\delta}\|$$

$$\leq 2M\|\widetilde{\boldsymbol{\delta}} - \boldsymbol{\delta}\| + \frac{1}{\epsilon^2}(\|\nabla\Phi(\mathbf{0})\| + M\epsilon) \cdot \left(\|\boldsymbol{\delta}\| \cdot \|\boldsymbol{\delta}^\top - \widetilde{\boldsymbol{\delta}}^\top\| + \|\widetilde{\boldsymbol{\delta}}^\top\| \cdot \|\boldsymbol{\delta} - \widetilde{\boldsymbol{\delta}}\|\right)$$

$$\leq \left(4M + \frac{\|\nabla\Phi(\mathbf{0})\|}{\epsilon}\right) \cdot \|\widetilde{\boldsymbol{\delta}} - \boldsymbol{\delta}\|.$$

Thus, $\boldsymbol{F}(\boldsymbol{\delta})$ is locally Lipshitz for $\boldsymbol{\delta} \in \mathbb{B}(\mathbf{0}, \epsilon)$. According to Lemma D.1, the ODE (3) has a unique solution $\boldsymbol{\delta}(t) \in C^1([0, \infty); \mathbb{R}^d)$. $\qquad\square$

### D.4 LEMMA D.2

**Lemma D.2.** *Suppose $\varphi(t, \boldsymbol{\delta}, \eta)$ in (5) is locally Lipshitz for all $\boldsymbol{\delta}$ in a domain $D \subset \mathbb{R}^d$:*

$$\|\varphi(t, \widetilde{\boldsymbol{\delta}}, \eta) - \varphi(t, \boldsymbol{\delta}, \eta)\| \leq C_\varphi \|\widetilde{\boldsymbol{\delta}} - \boldsymbol{\delta}\|, \forall \widetilde{\boldsymbol{\delta}}, \boldsymbol{\delta} \in D. \tag{56}$$

*Let $W$ be a compact subset of $D$, $\boldsymbol{\delta}_0 \in D$, and suppose it is known that $\boldsymbol{\delta}_k$ generated by single-step method (5) and the solution $\boldsymbol{\delta}(t)$ of ODE (4) lie entirely in $W$, where $k = 1, 2, \cdots$ and $t \geq 0$. If the single-step method (5) has pth-order accuracy for solving ODE (4), when $\boldsymbol{\delta}_0 = \boldsymbol{\delta}(0)$, the global truncation error of $\boldsymbol{\delta}(t)$ at $t_k$ is*

$$\|\boldsymbol{\delta}_k - \boldsymbol{\delta}(t_k)\| = O(\eta^p),$$

*where $t_k = k\eta \leq T$ and $T \in [0, \infty)$.*

The proof follows papers (Kincaid et al., 2009; Sauer, 2011), and is included D.4 for completeness.

*Proof.* Suppose $\boldsymbol{\delta}_k = \boldsymbol{\delta}(t_k)$, based on (5),

$$\bar{\boldsymbol{\delta}}_{k+1} = \boldsymbol{\delta}(t_k) + \eta\varphi(t_k, \boldsymbol{\delta}(t_k), \eta). \tag{57}$$

Since the single-step method (5) has $p$th-order accuracy, there exists a constant $C > 0$,

$$\|\bar{\boldsymbol{\delta}}_{k+1} - \boldsymbol{\delta}(t_{k+1})\| \leq C\eta^{p+1}. \tag{58}$$

Combine (5) and (57),

$$\|\bar{\boldsymbol{\delta}}_{k+1} - \boldsymbol{\delta}_{k+1}\| \leq \|\boldsymbol{\delta}(t_k) - \boldsymbol{\delta}_k\| + \eta\|\varphi(t_k, \boldsymbol{\delta}(t_k), \eta) - \varphi(t_k, \boldsymbol{\delta}_k, \eta)\|. \tag{59}$$

From the condition, $W$ be a compact subset of $D$, $\boldsymbol{\delta}_0 \in D$, and it is known that $\boldsymbol{\delta}_k$ generated by single-step method (5) and the solution $\boldsymbol{\delta}(t)$ of ODE (4) lie entirely in $W$, where $k = 1, 2, \cdots$ and $t \geq 0$. Therefore, according to (56) and (59),

$$\|\bar{\boldsymbol{\delta}}_{k+1} - \boldsymbol{\delta}_{k+1}\| \leq (1 + \eta C_\varphi)\|\boldsymbol{\delta}(t_k) - \boldsymbol{\delta}_k\|. \tag{60}$$

Based on (58) and (60), the local truncation error of $\boldsymbol{\delta}(t)$ at $t_k$ is

$$\begin{aligned}
\|\boldsymbol{\delta}_{k+1} - \boldsymbol{\delta}(t_{k+1})\| &\leq \|\boldsymbol{\delta}(t_{k+1}) - \bar{\boldsymbol{\delta}}_{k+1}\| + \|\boldsymbol{\delta}_{k+1} - \bar{\boldsymbol{\delta}}_{k+1}\| \\
&\leq C\eta^{p+1} + (1 + \eta C_\varphi)\|\boldsymbol{\delta}(t_k) - \boldsymbol{\delta}_k\|.
\end{aligned}$$

Denote $e_k = \boldsymbol{\delta}_k - \boldsymbol{\delta}(t_k)$, we have

$$\|e_{k+1}\| \leq C\eta^{p+1} + (1 + \eta C_\varphi)\|e_k\|.$$

Then,

$$\begin{aligned}
\|e_k\| &\leq (1 + \eta C_\varphi)^k \|e_0\| + C\eta^{p+1} \sum_{i=0}^{k-1} (1 + \eta C_\varphi)^i \\
&\leq (1 + \eta C_\varphi)^k \|e_0\| + \frac{C\eta^{p+1}}{\eta C_\varphi} \left[(1 + \eta C_\varphi)^k - 1\right]
\end{aligned}$$

Since, $1 + \eta C_\varphi \leq e^{\eta C_\varphi}$, $(1 + \eta C_\varphi)^k \leq e^{k\eta C_\varphi}$, if $t_k = k\eta \leq T$, we have

$$\|e_k\| \leq \|e_0\| e^{TC_\varphi} + \frac{C\eta^p}{C_\varphi} (e^{TC_\varphi} - 1)$$

When $\boldsymbol{\delta}_0 = \boldsymbol{\delta}(0)$, $\|e_0\| = 0$ and

$$\|e_k\| \leq \frac{C\eta^p}{C_\varphi} (e^{TC_\varphi} - 1)$$

Therefore, the global truncation error of $\boldsymbol{\delta}(t)$ at $t_k$ is

$$\|\boldsymbol{\delta}_k - \boldsymbol{\delta}(t_k)\| = O(\eta^k)$$

$\square$

### D.5 PROOF OF THEOREM 4.3

*Theorem* 4.3. Suppose that (i) $\Phi(\boldsymbol{\delta}) \in \mathcal{F}_M$ and $\boldsymbol{\delta}^* \notin \mathbb{B}(\mathbf{0}, \epsilon)$, (ii) $\epsilon < \frac{\|\nabla\Phi(\mathbf{0})\|}{(U+1)M}$ with $U \geq 2$, (iii) $\boldsymbol{\delta}_0 \in \mathbb{S}(\mathbf{0}, \epsilon) \cap \mathbb{B}(\boldsymbol{\delta}^\triangle, \sqrt{2}\epsilon)$, as the step size $\eta \to 0$, PGD (2) converges to ODE (3) in the sense that, for all fixed $T > 0$,

$$\lim_{\eta \to 0} \max_{0 \leq k \leq \frac{T}{\eta}} \|\boldsymbol{\delta}_k - \boldsymbol{\delta}(k\eta)\| = 0 \tag{61}$$

*Proof.* Under conditions of 4.3, according to Lemma C.1, the trajectory of $\boldsymbol{\delta}_k$ generated by PGD (2) is on $\mathbb{S}(\mathbf{0}, \epsilon)$. Suppose $\boldsymbol{\delta}_k = \boldsymbol{\delta}(k\eta)$ and $t_k = k\eta$, based on (54), we have

$$\begin{aligned}
\boldsymbol{\delta}_{k+1} =& \boldsymbol{\delta}(t_k) + \eta \cdot \frac{\epsilon}{\|\boldsymbol{\delta}(t_k) - \eta\nabla\Phi(\boldsymbol{\delta}(t_k))\|} \cdot \\
& \left[ \boldsymbol{\delta}(t_k) \frac{\langle 2\boldsymbol{\delta}(t_k) - \eta\nabla\Phi(\boldsymbol{\delta}(t_k)), \nabla\Phi(\boldsymbol{\delta}(t_k)) \rangle}{\epsilon(\|\boldsymbol{\delta}(t_k)\| + \|\boldsymbol{\delta}(t_k) - \eta\nabla\Phi(\boldsymbol{\delta}(t_k))\|)} - \nabla\Phi(\boldsymbol{\delta}(t_k)) \right].
\end{aligned} \tag{62}$$

According to ODE (3), apply Taylor expansion for $\boldsymbol{\delta}(t_{k+1})$,

$$\begin{aligned}
\boldsymbol{\delta}(t_{k+1}) &= \boldsymbol{\delta}(t_k) + \eta \frac{\partial\boldsymbol{\delta}(t)}{\partial t}\Big|_{t=t_k} + I_d O(\eta^2) \\
&= \boldsymbol{\delta}(t_k) + \eta\left(\frac{1}{\epsilon^2}\boldsymbol{\delta}(t_k)\langle\boldsymbol{\delta}(t_k), \nabla\Phi(\boldsymbol{\delta}(t_k))\rangle - \nabla\Phi(\boldsymbol{\delta}(t_k))\right) + I_d O(\eta^2).
\end{aligned} \tag{63}$$

Combine (62) and (63) ,

$$\begin{aligned}
&\lim_{\eta \to 0} \frac{\|\boldsymbol{\delta}_{k+1} - \boldsymbol{\delta}(t_{k+1})\|}{\eta^2} \\
=& \lim_{\eta \to 0} \left\| \eta \cdot \frac{\epsilon}{\|\boldsymbol{\delta}(t_k) - \eta\nabla\Phi(\boldsymbol{\delta}(t_k))\|} \left( \boldsymbol{\delta}(t_k) \frac{\langle 2\boldsymbol{\delta}(t_k) - \eta\nabla\Phi(\boldsymbol{\delta}(t_k)), \nabla\Phi(\boldsymbol{\delta}(t_k)) \rangle}{\epsilon(\|\boldsymbol{\delta}(t_k)\| + \|\boldsymbol{\delta}(t_k) - \eta\nabla\Phi(\boldsymbol{\delta}(t_k))\|)} - \nabla\Phi(\boldsymbol{\delta}(t_k)) \right) \right. \\
& \left. - \eta\left(\frac{1}{\epsilon^2}\boldsymbol{\delta}(t_k)\langle\boldsymbol{\delta}(t_k), \nabla\Phi(\boldsymbol{\delta}(t_k))\rangle - \nabla\Phi(\boldsymbol{\delta}(t_k))\right) + I_d O(\eta^2) \right\| \Big/ \eta^2 \\
=& O(1).
\end{aligned}$$

Consequently, PGD method (2) exhibits first-order accuracy. In other words, if $\boldsymbol{\delta}_k = \boldsymbol{\delta}(t_k\eta)$,

$$\|\boldsymbol{\delta}_{k+1} - \boldsymbol{\delta}(t_{k+1})\| = O(\eta^2). \tag{64}$$

Based on formulations (5) and (54), we designate the incremental function as follows:

$$\varphi(\boldsymbol{\delta}, \eta) := \frac{\epsilon}{\eta} \cdot \frac{\boldsymbol{\delta} - \eta\nabla\Phi(\boldsymbol{\delta})}{\|\boldsymbol{\delta} - \eta\nabla\Phi(\boldsymbol{\delta})\|} - \frac{\boldsymbol{\delta}}{\eta}. \tag{65}$$

Because $\Phi(\boldsymbol{\delta}) \in \mathcal{F}_M$,

$$\|\nabla\Phi(\mathbf{0}) - \nabla\Phi(\boldsymbol{\delta})\| \leq M\|\boldsymbol{\delta}\| \Rightarrow \|\nabla\Phi(\mathbf{0})\| - M\epsilon \leq \|\nabla\Phi(\boldsymbol{\delta})\| \leq \|\nabla\Phi(\mathbf{0})\| + M\epsilon. \tag{66}$$

Then,

$$\epsilon \leq \|\boldsymbol{\delta} - \eta\nabla\Phi(\boldsymbol{\delta})\| \leq \|\boldsymbol{\delta}\| + \eta\|\nabla\Phi(\boldsymbol{\delta})\| \leq \epsilon + \eta(\|\nabla\Phi(\mathbf{0})\| + M\epsilon). \tag{67}$$

Combine (65), (66) and (67), for $\forall \boldsymbol{\delta}, \widetilde{\boldsymbol{\delta}} \in \mathbb{B}(\mathbf{0}, \epsilon)$, we have

$$\|\varphi(\widetilde{\boldsymbol{\delta}}, \eta) - \varphi(\boldsymbol{\delta}, \eta)\|$$

$$= \| (\frac{\epsilon}{\eta} \cdot \frac{\widetilde{\boldsymbol{\delta}} - \eta \nabla \Phi(\widetilde{\boldsymbol{\delta}})}{\|\widetilde{\boldsymbol{\delta}} - \eta \nabla \Phi(\widetilde{\boldsymbol{\delta}})\|} - \frac{\widetilde{\boldsymbol{\delta}}}{\eta}) - (\frac{\epsilon}{\eta} \cdot \frac{\boldsymbol{\delta} - \eta \nabla \Phi(\boldsymbol{\delta})}{\|\boldsymbol{\delta} - \eta \nabla \Phi(\boldsymbol{\delta})\|} - \frac{\boldsymbol{\delta}}{\eta}) \|$$

$$\leq \frac{\epsilon}{\eta} \cdot \left\| \frac{\|\boldsymbol{\delta} - \eta \nabla \Phi(\boldsymbol{\delta})\|(\widetilde{\boldsymbol{\delta}} - \eta \nabla \Phi(\widetilde{\boldsymbol{\delta}})) - \|\widetilde{\boldsymbol{\delta}} - \eta \nabla \Phi(\widetilde{\boldsymbol{\delta}})\|(\boldsymbol{\delta} - \eta \nabla \Phi(\boldsymbol{\delta}))}{\|\widetilde{\boldsymbol{\delta}} - \eta \nabla \Phi(\widetilde{\boldsymbol{\delta}})\| \cdot \|\boldsymbol{\delta} - \eta \nabla \Phi(\boldsymbol{\delta})\|} \right\| + \frac{1}{\eta}\|\widetilde{\boldsymbol{\delta}} - \boldsymbol{\delta}\|$$

$$\leq \frac{1}{\eta \epsilon} \cdot \left\| (\widetilde{\boldsymbol{\delta}} - \eta \nabla \Phi(\widetilde{\boldsymbol{\delta}}))\|\boldsymbol{\delta} - \eta \nabla \Phi(\boldsymbol{\delta})\| - (\boldsymbol{\delta} - \eta \nabla \Phi(\boldsymbol{\delta}))\|\boldsymbol{\delta} - \eta \nabla \Phi(\boldsymbol{\delta})\| + \right.$$

$$\left. (\boldsymbol{\delta} - \eta \nabla \Phi(\boldsymbol{\delta}))\|\boldsymbol{\delta} - \eta \nabla \Phi(\boldsymbol{\delta})\| - (\boldsymbol{\delta} - \eta \nabla \Phi(\boldsymbol{\delta}))\|\widetilde{\boldsymbol{\delta}} - \eta \nabla \Phi(\widetilde{\boldsymbol{\delta}})\| \right\| + \frac{1}{\eta}\|\widetilde{\boldsymbol{\delta}} - \boldsymbol{\delta}\|$$

$$\leq \frac{1}{\eta \epsilon} \cdot \left( \|\boldsymbol{\delta} - \eta \nabla \Phi(\boldsymbol{\delta})\| \cdot \|(\widetilde{\boldsymbol{\delta}} - \boldsymbol{\delta}) + \eta(\nabla \Phi(\boldsymbol{\delta}) - \nabla \Phi(\widetilde{\boldsymbol{\delta}}))\| + \|\boldsymbol{\delta} - \eta \nabla \Phi(\boldsymbol{\delta})\| \cdot \right.$$

$$\left. \left| \|\widetilde{\boldsymbol{\delta}} - \eta \nabla \Phi(\widetilde{\boldsymbol{\delta}})\| - \|\boldsymbol{\delta} - \eta \nabla \Phi(\boldsymbol{\delta})\| \right| \right) + \frac{1}{\eta}\|\widetilde{\boldsymbol{\delta}} - \boldsymbol{\delta}\|$$

$$\leq \frac{2}{\eta \epsilon} \cdot \left[ \epsilon + \eta(\|\nabla \Phi(\mathbf{0})\| + M\epsilon) \right] \cdot \left( \|\widetilde{\boldsymbol{\delta}} - \boldsymbol{\delta}\| + \eta M\|\widetilde{\boldsymbol{\delta}} - \boldsymbol{\delta}\| \right) + \frac{1}{\eta}\|\widetilde{\boldsymbol{\delta}} - \boldsymbol{\delta}\|$$

$$= C_\varphi \|\widetilde{\boldsymbol{\delta}} - \boldsymbol{\delta}\|, \tag{68}$$

where $C_\varphi = 2(\frac{1}{\eta} + \frac{\|\nabla \Phi(\mathbf{0})\|}{\epsilon} + M)(1 + \eta M) + \frac{1}{\eta}$. In accordance to (64) and (68), PGD method (2) demonstrates first-order accuracy, and the incremental function $\varphi(\boldsymbol{\delta}, \eta)$ complies with the Lipschitz condition for $\boldsymbol{\delta} \in \mathbb{B}(\mathbf{0}, \epsilon)$. In addition, Lemma C.1 and Theorem 4.1 guarantee that $\boldsymbol{\delta}_k$ generated by PGD method (2) and the solution $\boldsymbol{\delta}(t)$ of ODE (3) lie entirely in $\mathbb{B}(\mathbf{0}, \epsilon)$, where $k = 1, 2, \cdots$ and $t \geq 0$. Therefore, following Lemma D.2, $\forall T > 0$, we have

$$\|\boldsymbol{\delta}_k - \boldsymbol{\delta}(k\eta)\| = O(\eta),$$

and

$$\lim_{\eta \to 0} \max_{0 \leq k \leq \frac{T}{\eta}} \|\boldsymbol{\delta}_k - \boldsymbol{\delta}(k\eta)\| = 0.$$

$\square$

### D.6 Proof of Theorem 4.4

*Theorem 4.4.* Suppose that (i) $\Phi(\boldsymbol{\delta}) \in \mathcal{F}_M$ and $\boldsymbol{\delta}^* \notin \mathbb{B}(\mathbf{0}, \epsilon)$, (ii) and $\epsilon < \frac{\|\nabla \Phi(\mathbf{0})\|}{(U+1)M}$ with $U \geq 2$, (iii) $\boldsymbol{\delta}(0) = \boldsymbol{\delta}_0 \in \mathbb{S}(\mathbf{0}, \epsilon) \cap \mathbb{B}(\boldsymbol{\delta}^\triangle, \sqrt{2}\epsilon)$, if $\Phi(\boldsymbol{\delta})$ is twice continuously differentiable, we have that $\boldsymbol{\delta} = \boldsymbol{\delta}^\triangle$ is an exponentially stable equilibrium point of ODE (3). That is, there exist $r > 0$, $\beta > 0$, $C_4 < 0$ such that if $\|\boldsymbol{\delta}_0 - \boldsymbol{\delta}^\triangle\| < r$, the convergence rate of $\boldsymbol{\delta}(t)$ for finding $\boldsymbol{\delta}^\triangle$ is:

$$\|\boldsymbol{\delta}(t) - \boldsymbol{\delta}^\triangle\| \leq \beta e^{C_4 t}\|\boldsymbol{\delta}(0) - \boldsymbol{\delta}^\triangle\|,$$

where $\boldsymbol{\delta}(t)$ is the solution to ODE (3) with $t \geq 0$.

*Proof.* To establish the local linear convergence of PGD from a continuous-time perspective, we introduce the following lemma:

**Lemma D.3.** *(Khalil, 2002, Theorem 4.7) Let $\boldsymbol{\delta} = \boldsymbol{\delta}^\triangle$ be an equilibrium point for ODE (4), namely,*

$$\frac{\partial \boldsymbol{\delta}(t)}{\partial t} = \boldsymbol{F}(\boldsymbol{\delta}) \ with \ \boldsymbol{F}(\boldsymbol{\delta}^\triangle) = \mathbf{0},$$

*where $\boldsymbol{F} : \mathbb{D} \to \mathbb{R}^d$ is continuously differentiable and $\mathbb{D}$ is a neighborhood of $\boldsymbol{\delta}^\triangle$. Let*

$$A = \left. \frac{\partial \boldsymbol{F}(\boldsymbol{\delta})}{\partial \boldsymbol{\delta}} \right|_{\boldsymbol{\delta} = \boldsymbol{\delta}^\triangle},$$

*and $\chi$ denotes eigenvalue of $A$. If $\chi_i \in \mathbb{R}$ and $\chi_i < 0$, $i = 1, 2, \cdots, d$ for all eigenvalues of $A$, $\boldsymbol{\delta}^\triangle$ is an exponentially stable equilibrium point of ODE (4).*

Lemma D.3 is used to demonstrate that the optimal point $\boldsymbol{\delta}^\triangle$ is an exponentially stable equilibrium point of ODE (3).

According to Theorem 4.1, the conditions in Theorem 4.4 guarantee $\boldsymbol{\delta}(t) \subset \mathbb{S}(\boldsymbol{0}, \epsilon)$. Next, we prove that $\boldsymbol{\delta} = \boldsymbol{\delta}^\triangle$ is an exponentially stable equilibrium point for ODE (3). Denote

$$\boldsymbol{F}(\boldsymbol{\delta}) := \frac{1}{\epsilon^2}\boldsymbol{\delta}\langle\boldsymbol{\delta}, \nabla\Phi(\boldsymbol{\delta})\rangle - \nabla\Phi(\boldsymbol{\delta}). \tag{69}$$

Based on the analysis on **Case 2** in Section 2.1, $\nabla\Phi(\boldsymbol{\delta}^\triangle) = -\lambda\boldsymbol{\delta}^\triangle$ and

$$\boldsymbol{F}(\boldsymbol{\delta}^\triangle) = \frac{1}{\epsilon^2}\boldsymbol{\delta}^\triangle\langle\boldsymbol{\delta}^\triangle, \nabla\Phi(\boldsymbol{\delta}^\triangle)\rangle - \nabla\Phi(\boldsymbol{\delta}^\triangle) = -\frac{1}{\epsilon^2}\boldsymbol{\delta}^\triangle\langle\boldsymbol{\delta}^\triangle, \lambda\boldsymbol{\delta}^\triangle\rangle) + \lambda\boldsymbol{\delta}^\triangle$$

$$= -\lambda\boldsymbol{\delta}^\triangle + \lambda\boldsymbol{\delta}^\triangle = 0.$$

Thus, $\boldsymbol{\delta} = \boldsymbol{\delta}^\triangle$ is an equilibrium point for ODE (3). Based on (69),

$$\frac{\partial\boldsymbol{F}(\boldsymbol{\delta})}{\partial\boldsymbol{\delta}} = \frac{1}{\epsilon^2}\left[\frac{\partial\boldsymbol{\delta}}{\partial\boldsymbol{\delta}}\langle\boldsymbol{\delta}, \nabla\Phi(\boldsymbol{\delta})\rangle + \boldsymbol{\delta}(\frac{\partial\boldsymbol{\delta}}{\partial\boldsymbol{\delta}}\nabla\Phi(\boldsymbol{\delta}))^\top + \boldsymbol{\delta}(\nabla^2\Phi(\boldsymbol{\delta})\boldsymbol{\delta})^\top\right] - \nabla^2\Phi(\boldsymbol{\delta})$$

$$= \frac{1}{\epsilon^2}\left[E_d\langle\boldsymbol{\delta}, \nabla\Phi(\boldsymbol{\delta})\rangle + \boldsymbol{\delta}(E_d\nabla\Phi(\boldsymbol{\delta}))^\top + \boldsymbol{\delta}(\boldsymbol{\delta})^\top(\nabla^2\Phi(\boldsymbol{\delta}))^\top\right] - \nabla^2\Phi(\boldsymbol{\delta})$$

$$= \frac{1}{\epsilon^2}\left[E_d\langle\boldsymbol{\delta}, \nabla\Phi(\boldsymbol{\delta})\rangle + \boldsymbol{\delta}(E_d\nabla\Phi(\boldsymbol{\delta}))^\top + \boldsymbol{\delta}(\boldsymbol{\delta})^\top(\nabla^2\Phi(\boldsymbol{\delta}))^\top\right] - \nabla^2\Phi(\boldsymbol{\delta})$$

Therefore,

$$A = \frac{\partial\boldsymbol{F}(\boldsymbol{\delta})}{\partial\boldsymbol{\delta}}\bigg|_{\boldsymbol{\delta}=\boldsymbol{\delta}^\triangle}$$

$$= -\frac{1}{\epsilon^2}\left[E_d\langle\boldsymbol{\delta}^\triangle, \lambda\boldsymbol{\delta}^\triangle\rangle + \boldsymbol{\delta}^\triangle(\lambda\boldsymbol{\delta}^\triangle)^\top - \boldsymbol{\delta}^\triangle(\boldsymbol{\delta}^\triangle)^\top(\nabla^2\Phi(\boldsymbol{\delta}^\triangle))^\top\right] - \nabla^2\Phi(\boldsymbol{\delta}^\triangle)$$

$$= -\frac{1}{\epsilon^2}\left[\lambda\epsilon^2 E_d + \boldsymbol{\delta}^\triangle(\boldsymbol{\delta}^\triangle)^\top(\lambda E_d - \nabla^2\Phi(\boldsymbol{\delta}^\triangle))\right] - \nabla^2\Phi(\boldsymbol{\delta}^\triangle)$$

$$= -\frac{1}{\epsilon^2}\boldsymbol{\delta}^\triangle(\boldsymbol{\delta}^\triangle)^\top(\lambda E_d - \nabla^2\Phi(\boldsymbol{\delta}^\triangle)) + (-\lambda E_d - \nabla^2\Phi(\boldsymbol{\delta}^\triangle))$$

$$= (E_d - \frac{1}{\epsilon^2}\boldsymbol{\delta}^\triangle(\boldsymbol{\delta}^\triangle)^\top)(\lambda E_d - \nabla^2\Phi(\boldsymbol{\delta}^\triangle)) - 2\lambda E_d \tag{70}$$

To employ Lemma D.3, it is imperative to demonstrate that all eigenvalues of (70) are negative. Denote

$$M_1 = E_d - \frac{1}{\epsilon^2}\boldsymbol{\delta}^\triangle(\boldsymbol{\delta}^\triangle)^\top, \tag{71}$$

$$M_2 = \lambda E_d - \nabla^2\Phi(\boldsymbol{\delta}^\triangle). \tag{72}$$

Clearly, both $M_1$ and $M_2$ are Hermitian matrices. Because

$$\begin{vmatrix} E_d & \boldsymbol{\delta}^\triangle \\ (\boldsymbol{\delta}^\triangle)^\top & E_1 \end{vmatrix} = \begin{vmatrix} E_d & \boldsymbol{\delta}^\triangle \\ \boldsymbol{0} & E_1 - (\boldsymbol{\delta}^\triangle)^\top\boldsymbol{\delta}^\triangle \end{vmatrix} = \begin{vmatrix} E_d - \boldsymbol{\delta}^\triangle(\boldsymbol{\delta}^\triangle)^\top & \boldsymbol{0} \\ (\boldsymbol{\delta}^\triangle)^\top & E_1 \end{vmatrix},$$

we have

$$|E_d - \boldsymbol{\delta}^\triangle(\boldsymbol{\delta}^\triangle)^\top| = |E_1 - (\boldsymbol{\delta}^\triangle)^\top\boldsymbol{\delta}^\triangle|. \tag{73}$$

Apply (73) to (71),

$$|\chi E_d - \boldsymbol{\delta}^\triangle(\boldsymbol{\delta}^\triangle)^\top| = \chi^{d-1}|\chi - (\boldsymbol{\delta}^\triangle)^\top\boldsymbol{\delta}^\triangle| = \chi^{d-1}|\chi - \epsilon^2|,$$

Therefore, $\chi(\boldsymbol{\delta}^\triangle(\boldsymbol{\delta}^\triangle)^\top) = 0$ or $\epsilon^2$. Based on (71), $\chi(M_1) = 0$ or 1. Thus, $M_1$ is positive semidefinite. Based on (16) and (24),

$$\lambda > UM, U \geq 2. \tag{74}$$

According to (Nesterov, 2018), when $\Phi(\boldsymbol{\delta}) \in \mathcal{F}_M$ and $\Phi(\boldsymbol{\delta})$ is twice continuously differentiable,

$$ME_d \succeq \nabla^2\Phi(\boldsymbol{\delta}^\triangle) \succeq 0. \tag{75}$$

Apply (74) and (75) to (72),

$$M_2 \succeq UME_d - ME_d \succeq ME_d.$$

Therefore, $M_2$ is also positive semidefinite. Due to $\|M_1 M_2\| \le \|M_1\| \cdot \|M_2\|$, we have

$$\max \chi(M_1 M_2) \le \max \chi(M_1) \cdot \max \chi(M_2) < \lambda. \tag{76}$$

From (66), $\lambda = \frac{\|\nabla\Phi(\boldsymbol{\delta}^\triangle)\|}{\epsilon} \ge \frac{\|\nabla\Phi(0)\|}{\epsilon} - M > 0$. Combine (70) and (76),

$$\max \chi(A) \le \max \chi(M_1 M_2) - 2\lambda < \lambda - 2\lambda = -\lambda \le M - \frac{\|\nabla\Phi(0)\|}{\epsilon} < 0, \tag{77}$$

The eigenvalues of $A$ are all negative, namely, $\chi(A) < 0$. Thus, based on Lemma D.3, $\boldsymbol{\delta} = \boldsymbol{\delta}^\triangle$ is an exponentially stable equilibrium point for ODE (3). According to the definition of exponentially stable equilibrium point in the literature (Khalil, 2002), there exist $r > 0$, $\beta > 0$ and $C_4 < 0$ such that if $\|\boldsymbol{\delta}_0 - \boldsymbol{\delta}^\triangle\| < r$, the convergence rate of $\boldsymbol{\delta}(t)$ for finding $\boldsymbol{\delta}^\triangle$ is:

$$\|\boldsymbol{\delta}(t) - \boldsymbol{\delta}^\triangle\| \le \beta e^{C_4 t}\|\boldsymbol{\delta}(0) - \boldsymbol{\delta}^\triangle\|.$$

$\square$

# E ADDITIONAL EXPERIMENTS

## E.1 LINEAR FUNCTION

Expanding on the analysis provided in Sections 6.2, we continue to verify the local linear convergence of PGD with the convex and smooth objective function in both discrete-time and continuous-time scenarios. To achieve this, we employ PGD method (2) and ODE (3) to address the OP mentioned in Figure 1:

$$\min_{\boldsymbol{\delta} \in \mathbb{B}(\mathbf{0}, \epsilon)} \Phi(\boldsymbol{\delta}), \tag{78}$$

where $\Phi(\boldsymbol{\delta}) = \delta^{(1)} + \delta^{(2)}$ and $\boldsymbol{\delta} = (\delta^{(1)}, \delta^{(2)})^\top$. Then, $\Phi(\boldsymbol{\delta})$ has no equilibrium point $\boldsymbol{\delta}^*$, and the optimal point of (78) is $\boldsymbol{\delta}^\triangle = (-\frac{\epsilon\sqrt{2}}{2}, -\frac{\epsilon\sqrt{2}}{2})^\top$. Besides, $\Phi(\boldsymbol{\delta})$ is a linear function, so it is 0-smooth and convex, but not strongly convex. In the light of the conditions in Theorem 3.1,

$$\epsilon < \frac{\|\nabla\Phi((0,0)^\top)\|}{3M} = \frac{\|(1,1)^\top\|}{3 \times 0} = +\infty.$$

For simplicity, we set the constraint radius $\epsilon \in \{0.5, 0.1, 0.05, 0.02, 0.01, 0.005\}$ and the step size $\eta = 5 \times 10^{-5}$. $\boldsymbol{\delta}_0$ is initialized by drawing from a uniform distribution over $\{\boldsymbol{\delta}|\boldsymbol{\delta} \in \mathbb{S}(\mathbf{0}, \epsilon), \delta_1 < 0, \delta_2 < 0\}$.

From the discrete-time perspective, we directly apply PGD to solve OP (78), and the converging curves under various $\epsilon$ are illustrated in Figure 13. The results obtained from Figure 13 align with the results of Figure 6, thus substantiating our Theorem 3.1.

From the continuous-time perspective, utilizing ODE (3), we deduce the following ODE that corresponds to PGD for solving OP (78):

$$\frac{\partial\boldsymbol{\delta}(t)}{\partial t} = \frac{1}{\epsilon^2}(\delta^{(1)} + \delta^{(2)})\boldsymbol{\delta} - (1,1)^\top. \tag{79}$$

The convergent curves determined by ODE (79) under various $\epsilon$ are displayed in Figure 14. The findings from Figure 13 align with the results of Figure 7, thus affirming the analysis presented in Theorem 4.4.

It's important to note that Figures 13 and 14 are distinct: Figure 13 illustrates the convergence of $\boldsymbol{\delta}_k$ generated by the PGD in the discrete-time scenario, while Figure 14 illustrates the convergence of $\boldsymbol{\delta}(t)$ determined by ODE in the continuous-time scenario. Combining the analysis from both Figures 13 and 14, we arrive at the conclusion that PGD exhibits local linear convergence when the objective function is convex and smooth.

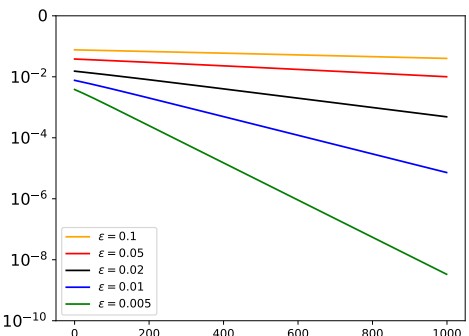

Figure 13: The convergence of PGD under different $\epsilon$ for solving OP (78). We use a logarithmic coordinate system where the horizontal axis represents the number of iterations $k$ and the vertical axis represents $\log_{10}(\|\boldsymbol{\delta}_k - \boldsymbol{\delta}^{\triangle}\|)$.

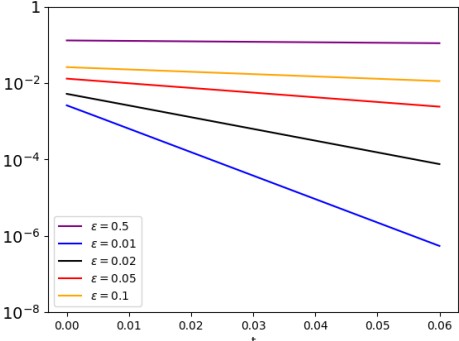

Figure 14: The convergence of $\boldsymbol{\delta}(t)$ decided by ODE (79) under different $\epsilon$ for solving OP (78). We use a logarithmic coordinate system where the horizontal axis represents evolutionary time $t$ and the vertical axis represents $\log_{10}(\|\boldsymbol{\delta}(t) - \boldsymbol{\delta}^{\triangle}\|)$.

### E.2 LOG-SUM-EXP FUNCTION

The PGD is applied to solve the constrained optimization problem

$$\min_{\boldsymbol{\delta} \in \mathbb{B}(\boldsymbol{0}, \epsilon)} \Phi(\boldsymbol{\delta}),$$

where $\Phi(\boldsymbol{\delta})$ represents the Log-Sum-Exp (used for constructing probability distributions):

$$\Phi(\boldsymbol{\delta}) = \log\left(\sum_{i=1}^{d} \exp(\delta_i)\right). \tag{80}$$

The log-Sum-Exp function (80) is convex but not strongly convex. No prior knowledge of the optimal solution $\boldsymbol{\delta}^{\triangle}$ is assumed, and the initialization is randomly sampled from the constraint ball $\mathbb{B}(\boldsymbol{0}, \epsilon)$.

We consider various dimensions $d \in \{50, 100, 200, 1000\}$ and constraint radius $\epsilon \in \{1, 5, 10, 20\}$. The experimental results for Log-Sum-Exp are shown in Figures 15. As illustrated, PGD demonstrates both global convergence and local linear convergence, which is consistent with our theoretical analysis.

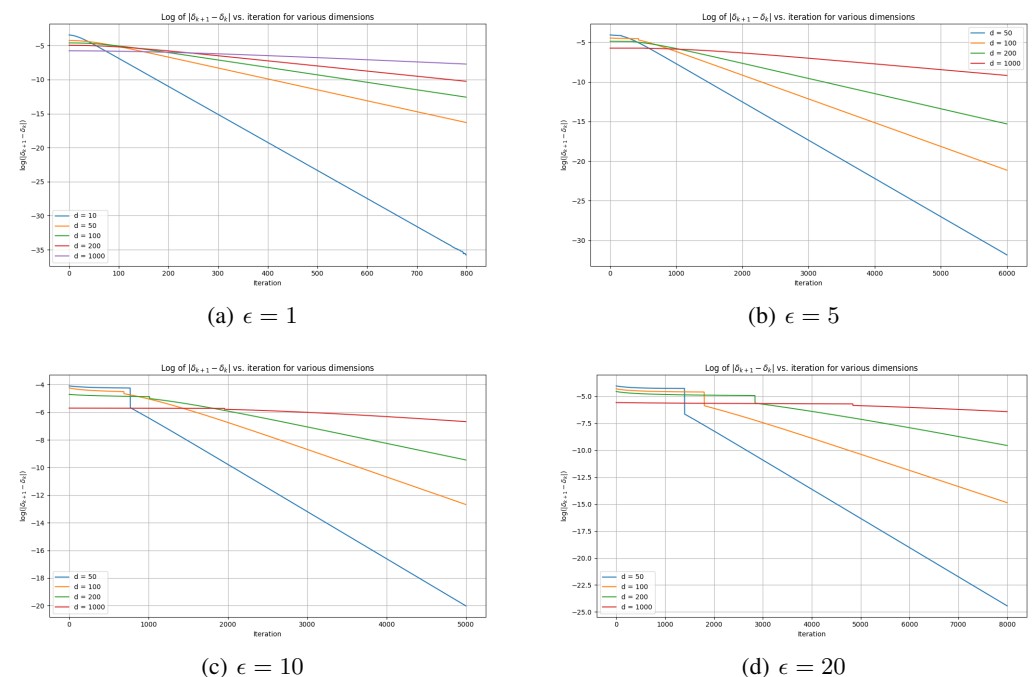

(a) $\epsilon = 1$         (b) $\epsilon = 5$

(c) $\epsilon = 10$         (d) $\epsilon = 20$

Figure 15: The convergence of $\boldsymbol{\delta}_k$ produced by PGD for solving optimization Log-Sum-Exp (80) with different dimensions $d$. We use a logarithmic coordinate system where the horizontal axis represents the number of iterations $k$ and the vertical axis represents $\log_{10}(\|\boldsymbol{\delta}_{k+1} - \boldsymbol{\delta}_k\|)$.

### E.3 LOGISTIC REGRESSION LOSS FUNCTION

Logistic Regression Loss (used for binary classification):

$$\Phi(\boldsymbol{\delta}) = \frac{1}{n} \sum_{i=1}^{n} \log\left(1 + \exp\left(-y_i \boldsymbol{w}^\top (\boldsymbol{x}_i + \boldsymbol{\delta})\right)\right), \tag{81}$$

where $\boldsymbol{w}$ is given vectors, $\boldsymbol{x}_i$ denotes the input features, and $y_i \in \{-1, 1\}$ are the labels. $\boldsymbol{w}$, $\boldsymbol{x}_i$ and $y_i$ are generated for simulation. This function is convex and smooth in $\boldsymbol{\delta}$, as it is a composition of the convex and smooth function $\log(1 + \exp(\cdot))$ with a linear transformation. The gradient does not vanish in the ball. Therefore, there is no critical point inside the interior of the constraint set.

The PGD is applied to solve the constrained optimization problem:

$$\min_{\boldsymbol{\delta} \in \mathbb{B}(\boldsymbol{0}, \epsilon)} \Phi(\boldsymbol{\delta}),$$

where $\Phi(\boldsymbol{\delta})$ represents the Logistic Regression Loss, which is convex but not strongly convex. No prior knowledge of the optimal solution $\boldsymbol{\delta}^\triangle$ is assumed, and the initialization is randomly sampled from the constraint ball $\mathbb{B}(\boldsymbol{0}, \epsilon)$.

We consider various dimensions $d \in \{10, 50, 100, 200\}$ and constraint radius $\epsilon \in \{0.1, 0.5, 1, 5\}$. The experimental results for Logistic Regression Loss are shown in Figures 16. As illustrated, PGD demonstrates both global convergence and local linear convergence, which is consistent with our theoretical analysis.

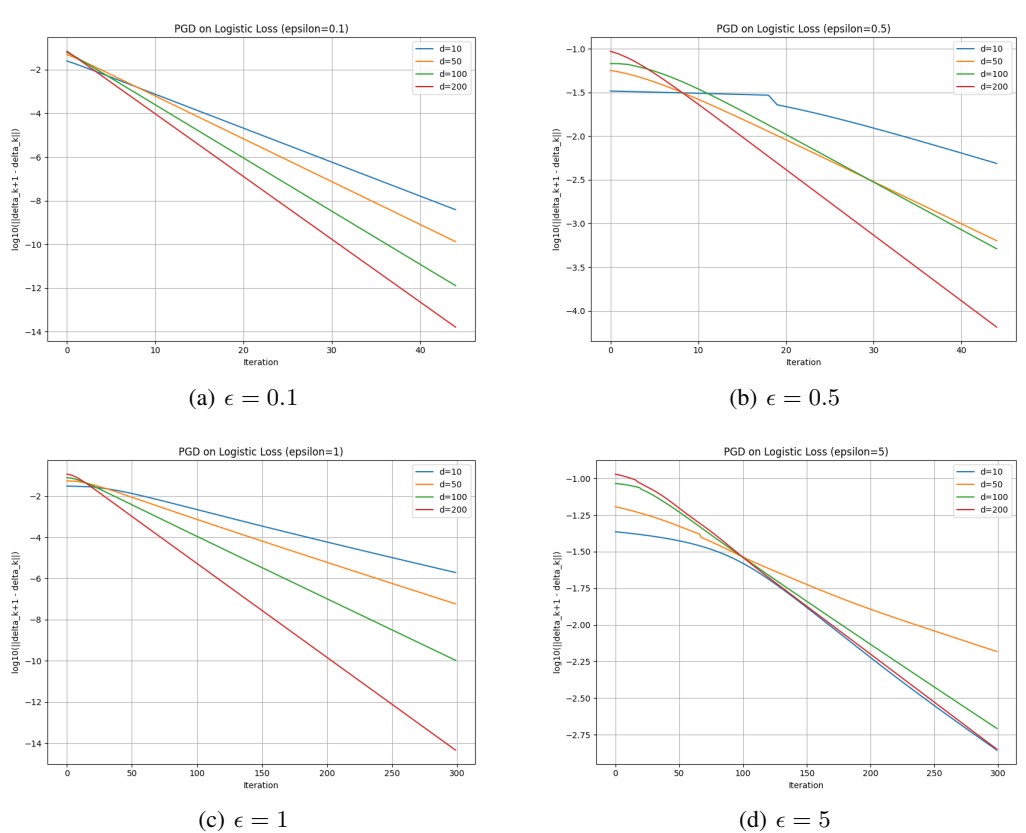

(a) $\epsilon = 0.1$

(b) $\epsilon = 0.5$

(c) $\epsilon = 1$

(d) $\epsilon = 5$

Figure 16: The convergence of $\boldsymbol{\delta}_k$ produced by PGD for solving optimization Logistic Regression Loss (81) with different dimensions $d$. We use a logarithmic coordinate system where the horizontal axis represents the number of iterations $k$ and the vertical axis represents $\log_{10}(\|\boldsymbol{\delta}_{k+1} - \boldsymbol{\delta}_k\|)$.

