# OpenReview forum: "Local Linear Convergence of Projected Gradient Descent: A Discrete and Continuous Analysis"
_ICLR.cc/2026/Conference — ICLR 2026 Conference Withdrawn Submission_

### Official Review · Reviewer_pCq3 · 2025-10-28

**Soundness:** 3
**Presentation:** 2
**Contribution:** 1
**Rating:** 2
**Confidence:** 4

**Summary:**

The paper studies projected gradient descent (PGD) under Euclidean ball constraints. It proves linear convergence under appropriate initialization and derives an ordinary differential equation (ODE) with trajectories approximating the discrete dynamics.

**Strengths:**

The paper derives an ODE for optimization problems on the sphere, strengthening the connection between continuous-time and discrete optimization.

**Weaknesses:**

1) The experimental section is minimal: plots that compare convergence and toy benchmarks (linear, log-sum-exp, logistic regression), with unspecified datasets. Larger experiments are required.

2) For smooth convex functions with minimizer lying outside of the considered compact set $\mathcal{C}$, functions might be strongly-convex on this given compact. For instance, if the function is twice-differentiable, and
$$\lambda_{\min}\left(\nabla^2 f(x)\right) \geq c > 0,$$
we might take
$$\mu = \min_x \lambda_{\min}\left(\nabla^2f(x)\right).$$
Consequently, linear convergence of projected gradient descent can hold locally on such compact subsets, not only for the case of ball constraints. Therefore, linear convergence under the Euclidean constraints is an incremental result.

3) The paper lacks a thorough comparison with prior work on projected gradient methods.

**Questions:**

1) How does the fast-convergence theory extend to constraint sets other than Euclidean balls?
2) Theorems 3.1–4.3 require the initial point to lie in $\mathbb{B}(\delta^{\Delta},\,\varepsilon\sqrt{2})$. What fraction of the initial ball’s volume does this region occupy as the dimension grows?
3) If PGD is initialized arbitrarily within the ball, how many iterations are needed before it enters the linear-convergence region?
4) How can the constant $C_4$ in Theorem 4.4 be computed (or tightly bounded) in practice?

**Details Of Ethics Concerns:**

No additional ethical concerns.

---

### Official Review · Reviewer_bMVv · 2025-10-31

**Soundness:** 3
**Presentation:** 2
**Contribution:** 3
**Rating:** 6
**Confidence:** 3

**Summary:**

This paper studies the projected gradient descent (PGD) algorithm, where the projection occurs on $\mathbb{B}(0,\varepsilon)$, applied to convex functions with $M$-Lipschitz gradients. They show that, locally, it exhibits linear convergence, which contrasts with the usual sublinear convergence speed of gradient descent for convex functions. The authors also introduce an ODE, that is the limit with vanishing stepsize of the studied PGD algorithm. The local linear convergence property also occurs for the solution of this ODE. Finally, experiments are conducted to validate that (i) the ODE accurately models PGD and (ii) the local linear convergence property is observed empirically.

**Strengths:**

I believe the contributions are interesting, and may challenge an usual intuition about the convergence of first-order algorithms for convex functions. The derived ODE, continuous equivalent of PGD, seems to be new, and offers interesting perspective to further study this problem. Finally, the paper is overall clear, and the introductory figures effectively illustrate the key concepts.

**Weaknesses:**

- I suggest to add a paragraph, at least in the "related works" appendix, about existing local convergence results for PGD.
- The purpose of Section 5 is not clear to me. It looks like a summary of the results of Section 3 and 4. Am I missing something ? Otherwise, I suggest to delete or at least reduce this section, and maybe, accordingly to one of my question, to use the saved space to elaborate more on the discrete setting.
- Remark 3.2: "Theorem 3.1 remains valid even if $\Phi(\delta)$ doesn’t have $\delta^\ast$". I think this sentence should be clarified.


Minor remark:
- For ease of reading, I suggest to recall in Section D the expression of ODE (3).
- There is a typo in Section 6.2.2: "PGD exhabits linear convergence rate"

**Questions:**

- I am surprised that the linear rates of Theorem 3.1 do not depend of the smoothness constant $M$, which is to my knowledge rather unusual. Do you have  an intuitive explanation about why it is so ?
- I believe it is a bit weird that in the main text, while the continuous section is almost two pages long, the discrete section is only half a page long. In particular, the fact that the local linear convergence property of Theorem 3.1 can be extended to a more general neighborhood is only stated as a remark, and deferred in appendix. In the end, if the continuous perspective can offer interesting insights, discrete results are the one we can use in practice. Could you please explain the reason for your choice?

---

### Official Review · Reviewer_DKPi · 2025-10-31

**Soundness:** 2
**Presentation:** 1
**Contribution:** 1
**Rating:** 2
**Confidence:** 4

**Summary:**

This work considers the minimization of a convex and Lipschitz smooth function over a ball and analyzes the Projected Gradient Descent (PGD) method for such problems. The main results consist of (1) deriving a continuous-time ODE that describes the limiting behavior of the PGD iterations as the step size goes to 0; and (2) establishing a local linear convergence rate of PGD, both in the discrete-time and continuous-time setting.

**Strengths:**

The geometric perspective behind the proof techniques seems original, although quite specific.

I didn’t check the proofs in detail, but the local linear convergence rate result for PGD on problem (1) under the given assumptions is correct. See also *Weaknesses*.

**Weaknesses:**

The authors seem unaware of the importance of quadratic growth conditions in the context of linear convergence rates. Relevant literature is missing (e.g., D. Drusvyatskiy and A. Lewis, Error Bounds, Quadratic Growth, and Linear Convergence of Proximal Methods) or not adequately discussed (e.g., (Necoara et al., 2019) is mentioned in the context of *sublinear convergence*, whereas it establishes *linear convergence* under `quadratic functional growth’).

More importantly, the main result on local linear convergence becomes somewhat trivial if familiar with quadratic growth conditions. The assumption that the unconstrained minimizer(s) is (are) *outside* the ball ensures that a quadratic growth condition holds. Indeed, the optimality conditions read $-\nabla \Phi(\delta^{\bigtriangleup}) = \lambda \delta$ for some multiplier $\lambda \geq 0$. If $\delta^\bigtriangleup \notin \mathbb{B}(0, \epsilon)$, this implies that $\nabla \Phi(\delta^{\bigtriangleup}) \neq 0$ and hence $\lambda > 0$. By convexity of $\Phi$ it follows that the Lagrangian is strongly convex around $\delta^\bigtriangleup$, and this in turn implies that $\Phi$ satisfies a quadratic growth condition.

As stated by the authors, the considered problem is very specific (ball constraint).

**Questions:**

Suggestions / Remarks

The citations in this work are not always accurate. For example, when listing methods like interior point, Lagrange multiplier or even PGD methods, it would be better to cite classical works, rather than some recent works. Besides, when the constraint is a simple ball, it arguably does not make a lot of sense to use interior point or Lagrange multiplier methods.

The function $F$ in equation (4) is not defined in the main text. The required properties (e.g., in Lemma D.1) deserve to be discussed, I believe.

At lines 38-40 the authors mention *the* equilibrium (stationary) point of $\Phi$. A convex function does not necessarily have a unique minimizer. This entails an additional assumption and should be stated as such.

---

### Official Review · Reviewer_QTYh · 2025-11-01

**Soundness:** 3
**Presentation:** 3
**Contribution:** 3
**Rating:** 6
**Confidence:** 3

**Summary:**

**Summary**

This paper establishes the local linear convergence of projected gradient descent applied to a smooth convex function $f$ when the constraint set is a Euclidean ball and the unconstrained minimizer of $f$ falls outside the constraint set. The authors conduct analysis in both discrete time and continuous time scenarios. Some experiments validate the theoretical findings.

**Strengths:**

**Strength**
The paper is well-written and easy to follow. It contains rich geometric intuitions and provides nice new insights into the local behavior of PGD. The idea of leveraging the geometry of the constraint to achieve better convergence behavior is interesting.

**Weaknesses:**

**Weaknesses**

1. Unclear explicit local linear convergence region

   **Theorem 3.1**  shows the existence of a local linear convergence region. However, the dependence on parameter $\theta$ is not adequately discussed. How does $\theta$ depend on the other problem parameters and the initial point? I believe the result would be even more interesting if an explicit characterization of $\theta$ is available.

2. Rigor and clarity of the proofs

   I appreciate the rich geometric intuitions behind the proofs. However, some proof arguments are based on figure (e.g., line 831) and lack rigor. I would also suggest that the authors add step-by-step explanations to the important steps of the proof.

Overall, I find the paper contains interesting insights into the role played by constraint geometry in PGD. The paper is well-written and easy to follow. Given the time limit of the review process, I have run simulations and empirically verified the claims in the paper, but there's insufficient time to verify all the proof details.

For now, I recommend weak acceptance. And I'll raise my score to 8 if my questions are appropriately addressed.

**Questions:**

**Questions**

1. In **Theorem 3.1**, The dependence on $U$ seems to suggest it should be fixed at 2, and there is no trade-off. Is there any intuition here?
2. According to **Theorem 3.1**, there is a lower bound on the distance to the optimal solution. However, suppose $\Phi$ is a well-conditioned smooth convex function such that $1 - \frac{1}{\kappa} < C_3$. Then how would this lower bound reconcile with the upper bound of projected gradient descent?
3. Proofs between line 800 and 803 look unclear to me. Could you explain these three lines in detail?
4. Do you think it's possible to extend the analysis to other constraint sets (e.g. strongly convex set)?
5. The paper's main results essentially leverage the structure of the constraint set. And it is well-known that Frank-Wolfe method is also capable of leveraging constraint geometry. And it is known that under the same condition that the unconstrained optimum is outside the feasible region, Frank-Wolfe also exhibits linear convergence [1]. Could you elaborate on the possible connection between these two methods?

**Minor issues**

1. Line 40

   Could you use optimal solution instead of equilibrium point?

2. Line 60

   I don't see why superlinear convergence is plotted here.

3. Line 71

   Gradient descent cannot guarantee $1/K$ rate of distance to $\delta^\Delta$ in general.

4. Line 94

   The definition of $C_3$ seems ambiguous here.

5. Line 147

   Please be precise about "doesn't have $\delta^*$".

6. Line 152

   $F(\delta)$ is not defined here.

7. Line 163

   incremental function => incremental functions.

8. Line 183, 262, 270, 300

   Missing full stop.

9. Line 265

   "which states that ..." is not clear.

10. Line 759

    The second equality should not be aligned right below the first one.

11. Line 779

   The second $<$ should be $\leq$.

**References**

[1] Garber, D., & Hazan, E. (2015, June). Faster rates for the Frank-Wolfe method over strongly-convex sets. In *International Conference on Machine Learning* (pp. 541-549). PMLR.

**Details Of Ethics Concerns:**

N/A.

---

### Note · Authors · 2025-11-19

I have read and agree with the venue's withdrawal policy on behalf of myself and my co-authors.